# Chromatin states shaped by an epigenetic code confer regenerative potential to the mouse liver

Chi Zhang[1], Filippo Macchi [1], Elena Magnani[1] & Kirsten C. Sadler [1✉]

We hypothesized that the highly controlled pattern of gene expression that is essential for liver regeneration is encoded by an epigenetic code set in quiescent hepatocytes. Here we report that epigenetic and transcriptomic profiling of quiescent and regenerating mouse livers define chromatin states that dictate gene expression and transposon repression. We integrate ATACseq and DNA methylation profiling with ChIPseq for the histone marks H3K4me3, H3K27me3 and H3K9me3 and the histone variant H2AZ to identify 6 chromatin states with distinct functional characteristics. We show that genes involved in proliferation reside in active states, but are marked with H3K27me3 and silenced in quiescent livers. We find that during regeneration, H3K27me3 is depleted from their promoters, facilitating their dynamic expression. These findings demonstrate that hepatic chromatin states in quiescent livers predict gene expression and that pro-regenerative genes are maintained in active chromatin states, but are restrained by H3K27me3, permitting a rapid and synchronized response during regeneration.

[1] Biology Program, NYU Abu Dhabi, Abu Dhabi, United Arab Emirates. ✉email: Kirsten.edepli@nyu.edu

Gene expression is largely dependent on the combinatorial influence of transcriptional regulators and a highly complex epigenetic code comprised of histone post-translational modifications (hPTMs), histone variants, DNA modifications, long noncoding RNAs, chromatin remodelers, and other factors[1]. In many cases, a combination of epigenetic marks demarcates distinct regions of the genome. The combined influence of these factors confers functions that facilitate precise regulation of gene expression, suppression of transposons and maintenance of nuclear structure. In simplistic terms, elements that pose a threat to cell identity or viability are packaged into constitutive heterochromatin, while other factors assemble on regions of open chromatin to maintain a permissive state for gene expression, DNA replication and recombination. Despite extensive work to study the interplay between the epigenetic features that organize the genome into these distinct chromatin states, defining how such states regulate essential transcriptional responses for organ homeostasis and regeneration is not yet known.

Tissue regeneration in response to injury or tissue loss is accompanied by widespread changes in gene expression and there is an emerging understanding that the regenerative transcriptome is, in part, due to epigenetic changes in regenerating tissues[1–3]. We hypothesized that an epigenetic code in quiescent tissue dictates the pattern of gene expression required for regeneration[3]. This is supported by studies in models where tissue regeneration is mediated by stem cells. For instance, the proliferative capacity and hence regenerative potential of stem cells in both skin[4] and intestines[5] is maintained by the polycomb repressor complexes (PRC), which mediate the repressive mark, trimethylated Histone H3 lysine 27 (H3K27me3)[6–9], which is part of a pattern that permits the precise expression of genes required for regeneration. In contrast, liver regeneration in mammals is largely accomplished by the proliferation of quiescent, mature hepatocytes which synchronously re-enter the cell cycle[10] following loss of liver mass. This process has been studied for over a century using the partial hepatectomy (PH) model in rodents[11], where restoration of liver mass relies on synchronous hepatocyte proliferation and hypertrophy, accompanied by changes in expression of thousands of genes[2,12–15]. Such coordinated changes in gene expression implicates that the epigenetic landscape facilitates the precise regulation of this process. However, only a few studies[1,2,16–19] have provided functional analysis of the epigenetic landscape in the liver.

The genome has been traditionally divided into repressed heterochromatin and open/active euchromatin. Active regions of the genome are dynamic, differ across cell types, and are marked by histone modifications that allow the chromatin to remain accessible to transcription factors. Heterochromatin is characterized by DNA methylation, repressive histone modifications such as H3K9me2/3 or H3K27me3, and association with the nuclear lamina. These epigenetic features of heterochromatin and small-RNA mediated mechanisms serve the essential role of suppressing transposable elements (TEs) to reduce the threat of their expression and mobilization in the genome[20–24]. The discovery of a third category of chromatin characterized as bivalently marked with both activating and repressive marks[25,26] reveals a mechanism to keep genes silenced under conditions where they are not needed but concurrently maintained in a poised state ready for activation in response to stimuli. Bivalent genes have primarily been studied in the context of development and cancer[27]. This highly complex epigenetic code, including the co-existence of marks that serve opposing functions in chromatin regulation, can functionally segment the genome into distinct regions. Our work addresses how such chromatin states regulate the complex gene expression pattern that confers the unique regenerative potential to the mammalian liver.

Genome-wide epigenetic profiling of hPTMs and DNA methylation combined with chromatin accessibility provided by ATACseq[28] provides a powerful tool to computationally identify regions of the genome with recurrent associations between marks. The collaborative ENCODE and Roadmap projects[29–31] and individual investigators have generated extensive datasets to define the epigenetic landscape of stem cells, cancer cells, developing embryos, and many normal tissue types. These studies uncovered common patterns of co-occurring epigenetic marks in many cell types, showing that active and repressed chromatin share similar marks in a wide variety of tissues. This has been extended by a systems biology approach that integrates multiple epigenetic marks to identify marks that co-exist at multiple sites in the genome. These are defined as chromatin states[32–34]. Each chromatin state shows specific enrichments in functional annotations, sequence motifs and genes serving shared cellular functions, suggesting distinct biological roles. In a pioneering study using a multivariate Hidden Markov Model, 51 distinct chromatin states were defined in human T cells[34]. This tool, ChromHMM, was expanded to find the chromatin state of multiple different cell types, and provided a robust method to define distinct epigenetic signatures of cell identities[30,32]. In addition to defining cell identity, chromatin states can be integrated with gene expression data to generate an understanding of the commonalities and differences underlying cell-type-specific regulation of gene expression. A recent report from the mouse ENCODE project combined 8 hPTMs across 66 tissue-stages during mouse development to define 15 ChromHMM states, of which most are consistent across tissues and stages, with the exception of the enhancer state, which is highly dynamic[35].

In contrast to hPTMs, DNA methylation on CpGs is relatively static and is enriched in gene bodies and in TEs[3,29,36–40]. DNA methylation has been shown to be a conserved mechanism to suppress TE expression[20,37,41–44] and thereby prevents the potentially catastrophic consequences for genomic stability posed by widespread TE mobilization. Since most TEs are fixed in the genome, the key role that DNA methylation plays in TE suppression in nearly all cells in mammals results in a consistent pattern of DNA methylation in regions of heterochromatin across diverse cell types, although some species-specific and cell-type differences and cell to cell variation in this pattern have been reported[19,36–40]. Constitutive heterochromatin is also marked by H3K9me3, and extensive crosstalk between DNA methylation and H3K9me3 serves to silence TEs and repress gene expression[45,46]. Yet, a simplistic model assuming that all TEs are packaged into constitutive heterochromatin is challenged by studies showing that nearly half of all TEs in human cells fall into active chromatin states[47] and findings that TEs, like genes, have unique epigenetic signatures that control their expression[2,23,47–50]. It is not clear how the epigenetic landscape serves to package TEs and genes into distinct chromatin regions to control their expression.

Our previous studies on liver regeneration showed that mice lacking the epigenetic regulator, Uhrf1, in hepatocytes displayed widespread DNA hypomethylation and enhanced regeneration following PH. This was associated with premature activation of pro-regenerative genes which we hypothesized was caused by the redistribution of H3K27me3 from promoters to transposons which had lost DNA methylation due to Uhrf1 depletion[2]. These findings suggested that, in wild type livers, these pro-regenerative genes would be maintained in regions of open, active chromatin but would be repressed by H3K27me3, indicating a program that drives liver regeneration as controlled by an epigenetic code.

In this work, we tested this hypothesis by combining genome-wide profiles of histone modifications, a histone variant, DNA methylation and chromatin accessibility by ATAC-seq to generate an epigenetic map of the adult mouse liver. This defined six chromatin states: two states were characterized by open chromatin and marks of active genes; these were devoid of TEs and encompassed nearly all the genes that were expressed in physiological conditions. However, a subset of genes in these active states were also marked by H3K27me3 and repressed in quiescent livers, representing bivalent genes. Many of these genes were reactivated during liver regeneration accompanied by loss of H3K27me3, and single-cell analysis revealed a heterogenous pattern of activation in regenerating hepatocytes, extending findings by others[51]. TEs were sequestered in states categorized as heterochromatin, and we found that the repressive marks that occupy TEs differed according to transposon age. This indicates that the chromatin state in quiescent livers regulates the expression of genes that maintain hepatic function, has a distinct pattern of occupancy across transposons and poises hepatocytes for regeneration when the mitotic stimuli is provided by PH.

## Results

**Defining chromatin states in the mouse liver.** Deciphering epigenetic codes has largely focused on hPTMs and DNA methylation. In part, this is because of the great diversity of hPTMs, which provide a rich set of elements from which a code can be constructed. Moreover, hPTMs regulation is versatile, as these modifications can be transformed, added or removed by regulating the enzymes that write or erase them. Some hPTMs are widely used as a proxy to indicate active and repressed genes: histone H3 lysine 4 tri-methylation (H3K4me3) recruits nucleosome remodeling enzymes and histone acetyltransferases[52] and enables RNA polymerase function. H3K27me3 largely marks facultative heterochromatin to repress gene expression and H3K9me3 marks constitutive heterochromatin[46]. We hypothesized that a combination of these marks would define functional regions of the hepatic genome.

To test this, we first investigated ENCODE datasets profiling adult mouse livers[53] using the multivariate hidden Markov model embedded in ChromHMM[34] on the seven available ChIPseq datasets (POLR2A, H3K4me3, H3K27ac, H3K4me1, H3K79me3, H3K36me3, and H3K27me3). This identified six states (Supplementary Fig. 1A), which, in sum, covered 24.83% of the genome (Supplementary Fig. 1B). The majority of the hepatic genome (75.17%) was in E-S5, and was devoid of any of the profiled features (Supplementary Fig. 1A-B). We sought to extend the epigenomic profiling of the mouse liver by surveying some epigenetic marks of interest (i.e., H3K9me3) and to include chromatin accessibility data and histone variants such as H2A.Z, which have a significant effect on the chromatin landscape[54,55]. We generated ChIPseq to profile marks of active (H3K4me3 and H2A.Z) and repressive (H3K27me3, H3K9me3) chromatin and ATACseq to profile chromatin accessibility (Supplementary Table 1) and incorporated DNA methylome profiling using previously generated eRRBS datasets[2,39]. Cluster analysis was used to determine if these datasets could be combined with ENCODE data, revealing a strong batch effect (Supplementary Fig. 2A), which could be attributed to the higher sequencing depth in our samples (Supplementary Fig. 2B-C). We therefore proceeded to use the datasets we generated. We are cognizant that this tissue contains multiple cell types, each of which can have distinct epigenetic patterns, but since hepatocytes are the dominant cell type in the adult liver, we speculate that the hepatocyte epigenome dominates the signal in these datasets.

The four ChIPseq and the ATACseq datasets were integrated by modeling the combinatorial presence or absence of these signals using ChromHMM[34]. DNA methylation data were not included at this stage as the base-pair resolution format of eRRBS data across a portion of the genome is difficult to incorporate with peak-based data. We optimized ChromHMM to identify six distinct chromatin states (S1-S6; Fig. 1A). Three states (S1, S2, and S3) were enriched for marks of open chromatin, of which, only S1 and S2 were highly enriched for H3K4me3, while S3 also has indications of being euchromatic, but was dominated by H2A.Z (Fig. 1A). Two states (S5 and S6) were characterized by the repressive marks, H3K9me3 and H3K27me3, respectively, and were inaccessible based on the absence of ATACseq signal.

Notably, S4 was devoid of any of the epigenetic marks considered in this study. This state accounts for 88.5% of the total genome, whereas states representing open chromatin (S1–3) cover 4% of the genome and the closed chromatin states cover 7.5% of the genome (Fig. 1B). This distribution is very similar to findings reported by the Mouse ENCODE Consortium for a range of adult cell types[30,31,53]. To determine if the exclusion of the ENCODE data had a significant effect on the portion of the genome amenable to analysis, we combined the empty states from our dataset (S4) and the ENCODE dataset (E-S5). These almost entirely overlap, with only 21.5% of the regions in S4 and 7.2% of the regions in E-S5 as unique to each dataset (Supplementary Fig. 2D). This means that the prediction of chromatin states from our data is robust. This also confirmed the conclusion that regulatory elements, which are the focus of the marks profiled by ENCODE, only account for a small fraction of the mouse genome[30]. More recent ENCODE studies using an expanded series of eight datasets across tissues and developmental stages in mice showed that 33% of the genome could be assigned to a chromatin state in at least one tissue-stage sample[35]. Our comparison suggests that including ENCODE data provides a minimal enhancement of genome coverage (Supplementary Fig. 2D), and the significant difference between studies on the developing liver and our work on mature livers suggests that the embryonic epigenetic landscape is more dynamic than adult tissues and therefore more likely to have expanded functionally relevant domains.

S1 and S2 are enriched for ATACseq, H3K4me3 and H2A.Z, suggesting that actively transcribed genes would be found in these states. Genome browser views highlight these patterns: genes such as *Fbp1*, which is involved in gluconeogenesis and is highly expressed in hepatocytes are marked by open chromatin marks that characterize S1 (Fig. 1C). In contrast, S6 is dominated by H3K27me3, the dynamic repressive mark which suppresses Hox genes[56] (Fig. 1D). Meanwhile, H3K27me3 is also detected in S1 and S2, suggesting that there are regions of open chromatin in the mouse liver that are repressed by H3K27me3. The *Fbxo48* gene and *Eldr*, the EGFR lncRNA, are both expressed during mouse liver development and silenced in mature livers[35,53], representing such bivalent genes (Fig. 1E).

We combined DNA methylome analysis from four male mice with the other marks in pairwise correlations. eRRBS data are biased towards promoters, as this method enriches for CpG islands[39,57]. The CpGs found in accessible chromatin (ATACseq positive) or those in regions marked with H3K4me4 or H2A.Z were not methylated whereas nearly all the CpGs found in heterochromatin regions marked by H3K9me3 were methylated. The anti-correlation between DNA methylation and H2A.Z is consistent with findings from other systems, where H2A.Z is postulated to protect regions of the genome from DNA methylation[37,58–60]. Both methylated and unmethylated CpGs overlapped with H3K27me3 (Fig. 2A). There was a complex relationship between DNA methylation, chromatin state and CpG density: nearly all the CpGs in S4 and S5 are fully methylated

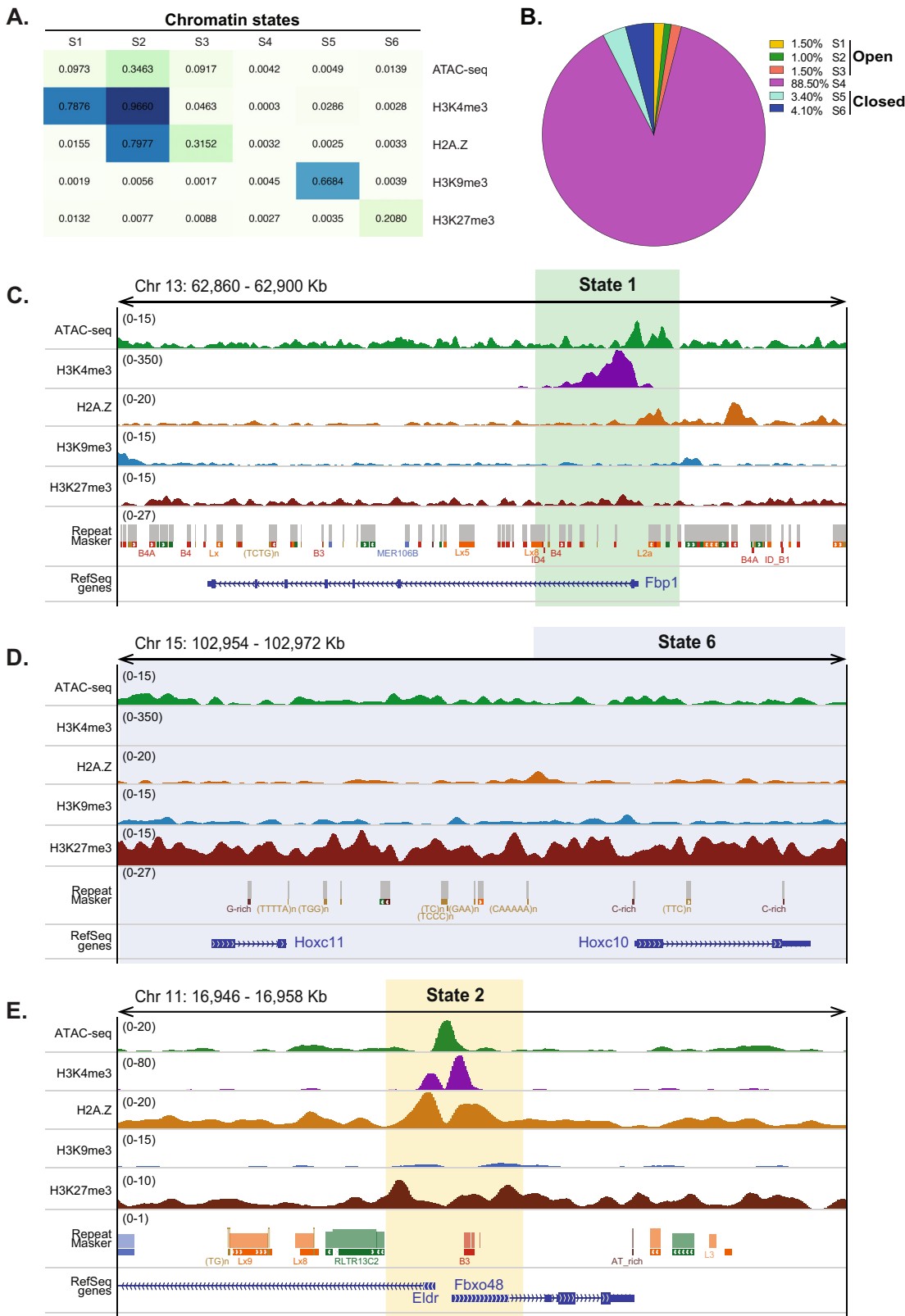

**Fig. 1 Epigenetic codes shape chromatin states in the adult mouse liver. A** Heatmap of the emission parameters for each mark profiled shows chromatin mark combinations associated with each chromatin state. Each column corresponds to a different state, and each row corresponds to a histone marker. The emission parameters were generated from ChIPseq data and represent the enriched possibility, indicated by values in each box. **B** The fraction of the hepatic genome covered by each state. The color of each state is retained throughout. **C–E** Representative genes showing the pattern of occupancy in on *Fbp1* (State 1), *Hox* genes (State 6), and a bivalent region from state 2 at the bottom.

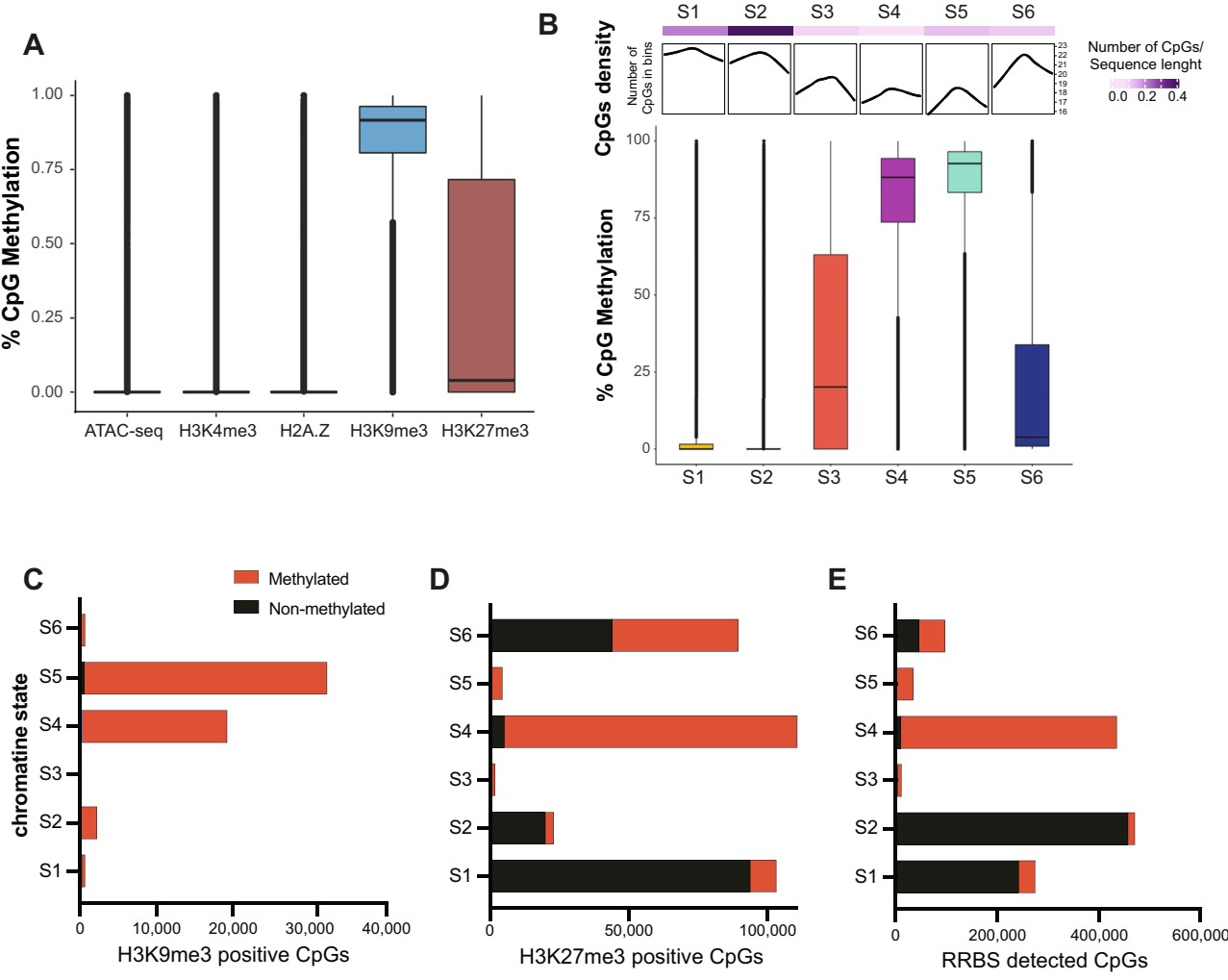

**Fig. 2 DNA methylation is enriched in repressive chromatin states and excluded from promoters. A** Methylation levels of individual CpGs in the regions designated as occupied by peaks from ATACseq and ChIPseq. The line in the boxplot represents the median of the distribution and boundaries of the boxes as the quartiles and the whiskers mark the 10th and 90th percentiles. **B** The CpG methylation status in each chromatin state is correlated with CpG density. The top panels display CpG density and the box and whisker plots display the range of CpG methylation across each state based on RRBS. The line in the boxplot represents the median and the boxes represent first and third quartiles, the whiskers represent data that are within the 1.5 × interquartile range; data beyond the end of the whiskers are outlying points that are plotted individually. **C–E**. Black indicates methylated, red indicates unmethylated. The number of CpGs categorized as methylated (>80%) and nonmethylated (<20%) plotted across (**C**) the entirety of each chromatin state or in the regions of each state occupied by H3K9me3 (**D**) or H3K27me3 (**E**).

(Fig. 2B), even though the CpG density was relatively low in these states. This is significant, since S4 occupies the majority of the genome (Fig. 1B). S5 is dominated by H3K9me3 and represents heterochromatin and, as expected, all the CpGs in this state are fully methylated (Fig. 2B). In contrast, despite high CpG density in S1 and S2, these were almost entirely unmethylated (Fig. 2B). This is consistent with S2 representing open and actively transcribed regions with unmethylated high-density CpGs regions, i.e., CpGs islands, which are enriched in promoters and are depleted of methylation[38,61].

To investigate the relationship between the individual chromatin marks and DNA methylation, we assessed the methylation profile of CpGs in regions marked by H3K9me3 (Fig. 2C) and H3K27me3 (Fig. 2D). Nearly all CpGs in regions covered by H3K9me3, regardless of the state, were fully methylated (Fig. 2C). However, we also found that not all methylated CpGs were occupied by H3K9me3, indicating a pattern whereby some highly methylated regions are in constitutive heterochromatin, whereas others are not. A more complex pattern was uncovered between

H3K27me3 and DNA methylation in S6, where the methylation status of CpGs covered by H3K27me3 ranges from 0 to 75%, with the median at 5% (Fig. 2A). This is, in part, attributed to the broad peak feature of H3K27me3 (Fig. 1D). A clear pattern was observed in S4, where all CpGs were methylated. In S1 and S2, nearly all the CpGs occupied by H3K27me3 were unmethylated (Fig. 2D), likely reflecting the general CpG methylation pattern in those states (Fig. 2E). Together, these data show that some features of the hepatic epigenome follow well-established patterns; for instance, well-established markers of heterochromatin have a high probability of co-occupancy with DNA methylation and that H2A.Z is anticorrelated to it[37,55,59,60]. Furthermore, we also uncovered unexpected patterns including the complex distribution of H3K27me3 across chromatin states and the bimodal pattern of DNA methylation in regions covered by H3K27me3.

**Chromatin states predict genomic elements and the transposon landscape.** If each chromatin state serves distinct functional roles

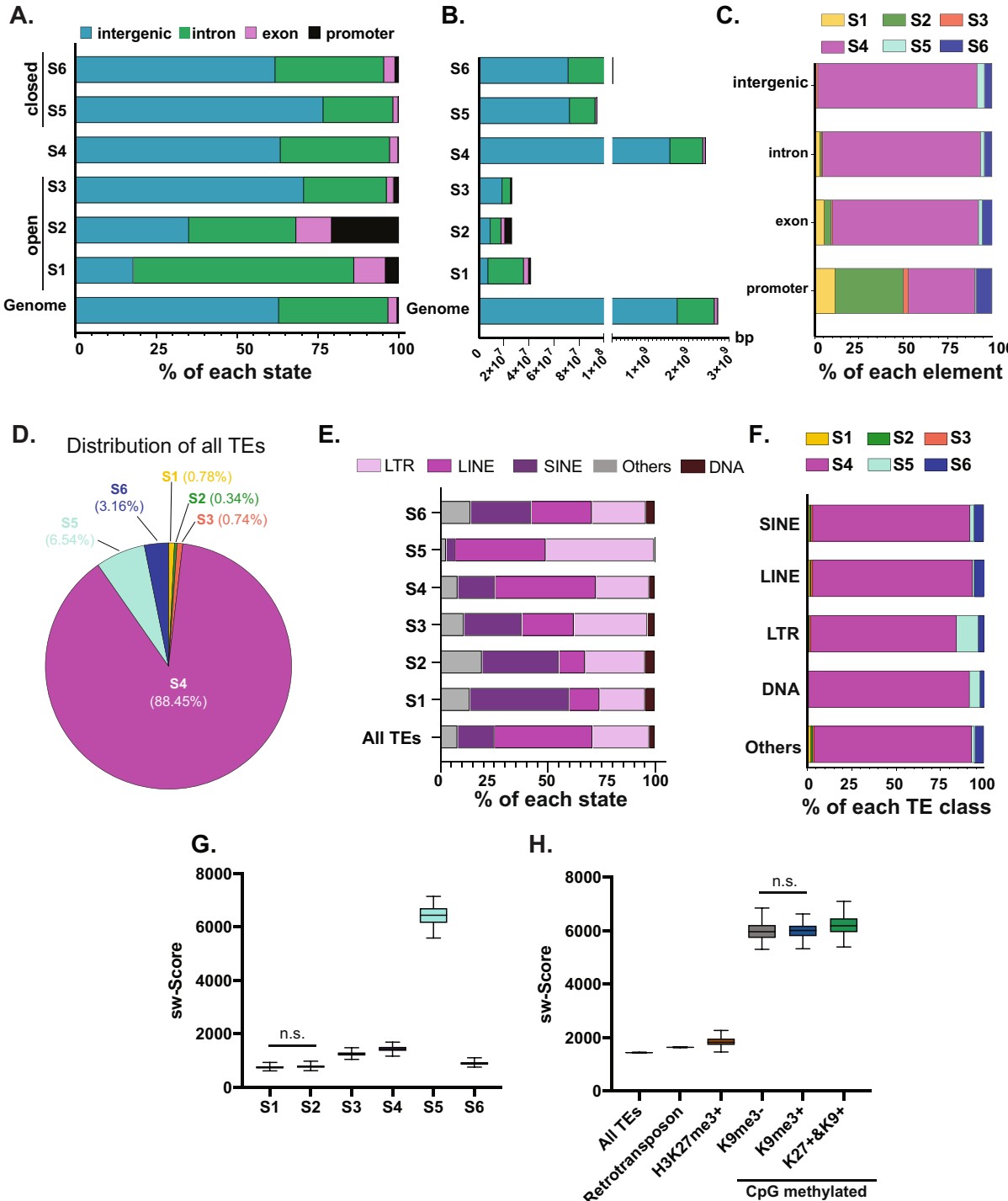

in genome organization, then the distribution of genomic elements across states should reflect this. For instance, since S1 and S2 are designated as open, active regions, this predicts that these states will largely be composed of promoters and gene-rich regions, whereas the heterochromatic S5 and the heavily methylated S4 would be predicted to be enriched in intergenic regions and introns. We tested this by categorizing the annotated introns, exons and intergenic regions and added proximal promoters as −500 bp from the TSS of each annotated gene (Fig. 3A-B) of each state. As predicted, S1 and S2 are enriched for promoters, with over half of all promoters in the genome occupied by S1, S2, and

S3 and only a small fraction of the intergenic regions is present in these open states (Fig. 3C). In contrast, S5 occupies less than 1% of all promoters (Fig. 3C) and is enriched for intergenic regions and introns (Fig. 3A-B), confirming its heterochromatin classification. S6, which is characterized by high probability of the presence of H3K27me3, covers nearly 9% of all promoters and is enriched for exons, but has relatively less occupancy of the intergenic regions compared to the total genome (Fig. 3B-C). This is consistent with a dynamic and highly complex role for H3K27me3 in gene regulation and genome organization in the liver.

**Fig. 3 The hepatic genomic and TE landscape is segregated by chromatin states. A** Relative proportion of annotated genomic elements across each chromatin state. **B** The length in base pairs (bp) covered in each chromatin state with the annotated genomic elements each individual state compared to the reference genome. Colors of each genomic element as labeled in legend for panel **A**. **C** Each genomic element is differentially occupied by the six predicted chromatin states. **D** Distribution of all TEs across chromatin states. **E** Annotation of TE classes and subclasses within each chromatin state compared to the distribution in the entire genome. **F** Distribution of chromatin states across TE classes and subclasses. **G** Smith-Waterman score represents the age of TEs in each chromatin state. The larger the sw score is, the younger the TE. sw score as a reflection of TE age is differentially distributed across states. One-way ANOVA is performed to test significant difference, multiple comparisons between each two group is with Tukey correction. The difference between S1 and S2 is not significant, all the other comparisons are significant with $p$-value < 0.0001. **H** All TEs, all retrotransposons and those TEs occupied by H3K27me3 alone or DNA methylation alone or in combination with H3K9me3 were assessed for their age based on sw score. Note that all TEs marked by H3K9me3 were also marked by CpG methylation, and therefore this mark could not be evaluated in isolation. One-way ANOVA is performed as in panel **G** The comparison between H3K9me3- and H3K9m3+ is marked with not significant, with all the other comparisons significant with $p$-value less than 0.0001. In **G, H**, the median is shown, with the boundaries of the boxes as the quartiles and the whiskers mark the 10th and 90th percentiles.

Studies in cultured cells have shown that there is variability in the epigenetic marks occupying TEs[23,48–50], with a simple model of TE repression mediated by packaging into constitutive heterochromatin, with DNA methylation as the key repressive mark. This predicts that TEs would be enriched in chromatin states that are marked by DNA methylation (S4) and H3K9me3 (S5). We assessed this by first determining the state where TEs reside and asked whether different TE classes were over or under represented in different states (Fig. 3D–F, Supplementary Fig. 3A). Most TEs (88%) reside in S4, 6.5% and 3.16% of all TEs are found in S5 and S6, respectively, and there are less than 2% of all TEs in S1–S3 combined (Fig. 3D, Supplementary Fig. 3A). This shows that while the distribution of TEs is proportional to the genome covered by S4, they are disproportionately distributed in the other chromatin states.

The majority TEs in mouse genome are class 1 retrotransposons, which are dominated by the LINE subclass (Fig. 3E). To investigate whether different chromatin states preferentially covered distinct TE classes and subclasses, their distribution across all states was examined (Fig. 3E-F, Supplementary Fig. 3A). The TE distribution in S4 was essentially the same as the whole genome, (Fig. 3E) which is logical since S4 encompasses 88% of the genome. LINEs were depleted from S1–S3, but enriched in S5. Less than 20% of all TEs are SINEs, yet these represented the largest category of TEs in S1, S2, and S6 and virtually no SINEs were found in S5 (Fig. 3E). S5 is selectively enriched with LTRs and LINEs (Fig. 3E-F), which suggest that these retroelements, which are the most likely to be mobile in the mouse genome, are sequestered in heterochromatin. Analysis of DNA methylation levels across TE classes (Supplementary Fig. 3B) and chromatin states (Supplementary Fig. 3C) shows that, as expected, all TEs have high levels of methylation compared to CpGs that are outside TEs; this is consistent with the high population of TEs in S4–6. In contrast, there was virtually no methylation on TEs in S2 (Supplementary Fig. 3C). This was, in part, reflective of low number of TEs in S2 (Supplementary Fig. 3A) and the low CpG density of the TEs in the open states (Supplementary Fig. 3D).

We hypothesized that multiple repressive mechanisms will be engaged to repress the young TEs, which pose the most threat of mobilization. The Smith-Waterman (SW) score provides a measure of TE age in each state. The SW score shows that TEs in S5 are the youngest (Fig. 3G). We asked whether younger TEs would be occupied by multiple epigenetic marks by comparing the SW score from all TEs marked with H3K27me3 alone, DNA methylation alone and DNA methylation in combination with H3K9me3 and H3K27me3 (Fig. 3H). We identified a significantly higher SW score in TEs marked by DNA methylation alone compared to all retrotransposons, and this score increased in those TEs with both DNA methylation and H3K9me3, but is not significantly different in TEs with all three repressive marks

(Fig. 3H). This corresponds to an elevated CpG density (Fig. 2B) in these younger TEs and the high level of methylation in the TEs marked by H3K9me3 (Supplementary Fig. 3E). This suggests that young retrotransposons may retain the CpG dense promoters found in the genomic structure of their retroviral ancestors.

**Chromatin states predict gene expression and functional categories**. To investigate the relationship between gene expression and hepatic chromatin state, the regions flanking the transcription start site (TSS; ±5 Kilobases (Kb)) of all genes in each state that contained any of the marks we profiled were identified (total = 16,940 genes; Fig. 4A, Supplementary Dataset 1). Nearly half of these (7778 genes) were in S2, and with the second highest percentage (27%; 4580 genes) in S4 (Fig. 4A). In contrast, there were less than 500 genes in S3 and S5, combined. This distribution is disproportionate with the percent of the genome covered by each state (Fig. 1B), indicating that S1 and S2 are significantly enriched for genes, and that S4 contains fewer genes than expected (see also Supplementary Fig. 6A). The enrichment of each epigenetic feature across the TSS of genes in each state (Fig. 4A) shows that in S2, the marks of active chromatin (ATACseq, H3K4me3, and H2A.Z) present strong and narrow peaks around the TSS, whereas S1 has fewer and broader peaks for ATACseq and H2A.Z. As expected, while all genes in S5 were covered with H3K27me3, very few also show occupancy by H3K4me3 and, strikingly, H3K27me3 was also enriched across the TSS of a subset of genes in S2 and S1. Since S4 occupies the majority of the genome and encompasses 4580 genes, we speculate that this region is not devoid of epigenetic marks, but instead is likely decorated with marks not included in our analysis. Indeed, although the heatmap in Fig. 4A is populated only with regions that contain a mark and therefore excludes the majority of the unmarked state, this data shows that some of the genes in this state are marked with H2A.Z and H3K4. Thus, although this state is marked as empty, there may be some genes, which are embedded in this broad swath of the genome marked by these features.

We tested the predictive value of these chromatin states on gene expression utilizing RNA-seq data from quiescent adult livers of the same age, gender, and strain of mice used for the epigenomic profiling[2]. Genes in each state were categorized as expressed based on Fragments Per Kilobase of transcript per Million (FPKM) > 1 (total = 9208 genes) or silenced (FPKM < 1; total = 7733 genes; Fig. 4B, Supplementary Dataset 1). Over ¾ of all genes in S2 and half of all genes in S1 were expressed, compared to only 6% of the genes in S6 (Fig. 4B). Despite the majority of the genome residing in S4, less than a third of the genes in this state were expressed, indicating that this state is largely repressive.

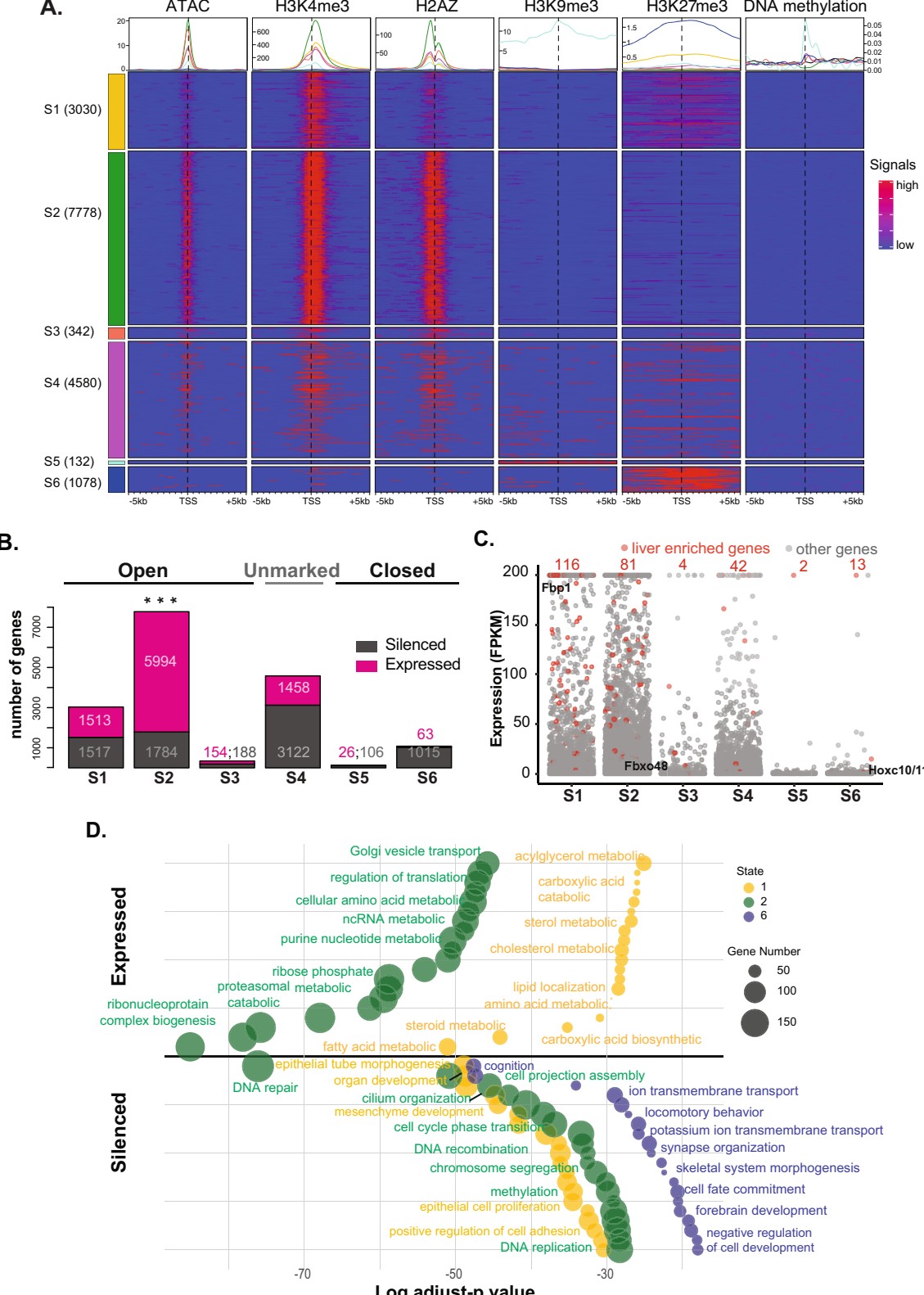

Chromatin states could serve as a mechanism for co-regulation of genes that share similar functional properties or that participate in the same cellular processes. This predicts that genes expressed at high levels in hepatocytes would be enriched in S1 and S2. We tested this by identifying a set of 242 genes that are expressed at high levels from a human liver-specific proteome dataset[62], termed as liver enriched genes (Supplementary Dataset 2). Plotting the expression levels of these genes compared to all genes in each state confirmed their high level of expression in the liver and also showed these genes to be significantly enriched in S1 and S2 and almost entirely absent from S3, S5, and S6 (Fig. 4C; red dots). This indicates that the chromatin state serves as a mechanism to maintain the constitutive high expression of genes important for liver identity and function.

**Fig. 4 Chromatin state predicts gene expression and function in the quiescent mouse liver. A** Global view of epigenetic features anchored around the TSS (±5 kb). Genes are segregated into chromatin states. The number of genes in each state is indicated. Genes in each state which were devoid of any marks were excluded from the display. Line colors in aggregate plots reflect the state colors as indicated on the side of the heatmap. **B** Numbers of expressed gene (FPKM > 1; pink) and silenced genes (FPKM ≤ 1; gray) in different states based on transcriptomic measurements in quiescent mouse livers. **C** Gene expression level, represented by FPKM, in each state. Liver enriched genes are indicated in red, and the numbers of the liver enriched genes in each state are labeled in red. The location of *Fbp1*, *Fbxo48*, and *Hoxc10/11* are indicated in alignment with data presented in Fig. 1. Liver enriched genes are indicated in red dots. **D** Significant GO pathways for genes in in S1, S2, and S6 which were designated as expressed and silenced in panel **B**. The bubble color and size correspond to state and gene number, respectively. The Y-axis displays arbitrary units used to spread the bubbles for ease of visualization.

To test whether certain cellular functions are embedded in the chromatin states identified, we categorized genes from each state as expressed and silenced (Fig. 4B) and subjected these groups to gene ontology (GO) analysis (Fig. 4D, Supplementary Fig. 4, Supplementary Dataset 3), revealing a striking pattern of genes with distinct functional features clustered by state. For instance, the expressed genes in S1 are all related to energy, lipid and amino acid metabolism while most of the expressed genes in S2 function in translation and nucleic acid metabolism (Fig. 4D). In contrast, there are no categories enriched for the expressed genes in S5 or S6, indicating that these states do not contain a code that functions to co-activate genes.

We were most intrigued by the silenced genes in each state. For instance, in S6, genes involved in functions unrelated to liver function, such as synapse organization, locomotion, and development, were silenced. This is consistent with the model whereby commitment to cell identity during liver development is, in part, due to H3K27me3 mediated silencing of genes that are not involved in differentiation or function of liver cells[63]. The genes in S4 captured diverse GO pathways (Supplementary Fig. 4), reflecting the absence of informative marks to cluster these genes into functional categories.

Our central hypothesis is that genes required for responding to stimuli are marked in quiescent livers by an epigenetic code that facilitates rapid response. This is exemplified by the genes that drive hepatocyte proliferation in response to loss of liver mass. The striking finding that many GO terms related to response to stimuli, such as the DNA damage response, cell cycle control and proliferation were enriched in the groups of silenced genes. also fall into states representing open chromatin (S1 and S2, Fig. 4D), support this hypothesis. Our previous study on epigenetic regulation of liver regeneration showed that H3K27me3 depletion from cell cycle genes is associated with premature activation of cell proliferation during liver regeneration[2]. Based on this, we hypothesized that H3K27me3 would be a key element of the epigenetic code coordinating gene expression during regeneration. To test this, we examined genes in S1 and S2 categorized based on expression and GO analysis (as in Fig. 4B, D), for H3K27me3 occupancy around the TSS in quiescent livers. This showed that silenced genes were enriched with H3K27me3 whereas expressed genes were not (Fig. 5a). Moreover, analysis of genes in S1 and S2 that were marked by H3K27me3, compared to those that were not marked, shows that the presence of H3K27me3 is correlated with significantly lower expression of these genes (p-value < 0.05, Supplementary Fig. 5A). Together, these data demonstrate that chromatin state is highly predictive of gene expression in the liver and that genes carrying out distinct functions can be clustered based on chromatin state. Most importantly, the subset of genes important for the response to stimuli are maintained in an active, open chromatin state but are repressed by H3K27me3 occupation.

**Bivalency primes pro-regenerative genes for expression during liver regeneration.** The discovery of bivalent genes marked with H3K27me3 and H3K4me3[26] illustrated that both activating and repressive marks can co-exist, and together these can influence important transcriptional events[27]. Since there is little hepatocyte proliferation under physiological conditions, it is not surprising that genes regulating the cell cycle and proliferation were silenced in the quiescent liver. Our discovery that these reside in active chromatin states (S1 and S2), suggests that the chromatin environment around these genes could contribute to their differential expression during regeneration.

To address this, we examined H3K27me3 during regeneration by two methods. First, western blotting for total H3K27me3 levels at 24, 30, 40, 96, 120, and 168 h and genome occupancy by ChIPseq at 30, 40, and 96 h after PH were performed (Supplementary Table 1). The total amount of H3K27me3 increased at 24 h and decrease at later timepoints (Fig. 5b). Although the global pattern of H3K27me3 genome occupancy did not dramatically differ between these timepoints, (Supplementary Fig. 5B), H3K27me3 occupancy at the TSS shows a loss at promoters at 30 and 40 h that is partially restored at 96 h (Supplementary Fig. 5C). The differences in the pattern of H3K27me3 genome occupancy is highlighted by comparing the number of peaks between quiescent livers and those at 30 h after PH: the total peak number is decreased at 30 h and only 42% of peaks at this timepoint occupy the same loci that are occupied in quiescent livers (Fig. 5c). Moreover, the peaks that are uniquely occupied by H3K27me3 in quiescent livers are primarily in exons and promoters, whereas the unique peaks at 30 h are mostly in introns and intergenic regions (Fig. 5d). Together, these data show a global decrease in the total amount of H3K27me3 and a redistribution during the stages of liver regeneration where hepatocytes are undergoing DNA replication and proceeding through mitosis. This could be mediated by changes in the levels of writers or erasers of H3K27me3, of which Ezh2, Eed, Suz12, Kdm6a, and Kdm6b show a dynamic expression pattern during regeneration (Supplementary Fig. 5D). In addition, other factors as of yet identified may be responsible for redistributing H3K27me3 away from promoters and towards the intergenic regions.

Notably, we found that of all the genes that were differentially expressed during regeneration, there was a significant enrichment for those residing in S2 (Supplementary Fig. 6A). This suggests an epigenetic code in S1 and S2 primes genes for differential expression during regeneration. This was further analyzed by focusing on all the genes in S1 and S2 that were silenced and were annotated by GO to be related to cell proliferation (Fig. 4D; Supplementary Dataset 2). These were grouped into three clusters by k-means algorithm based on H3K4me3 and H3K27me3 occupancy around the TSS at baseline, other timepoints after PH are sorted accordingly to clusters (Fig. 5e; Supplementary Dataset 2). Cluster 1 (109 genes) was highly enriched with H3K4me3 arounds the TSS with relatively lower H3K27me3 occupancy, Cluster 2 (207 genes) was enriched with H3K4me3 in downstream of the TSS and Cluster 3 (268 genes) had lower H3K4me3 occupancy but had relatively higher H3K27me3 occupancy. While genes in all clusters were significantly increased at 40 and 48 h after PH, Cluster 3 genes were the most repressed in the quiescent liver and are among the most active during

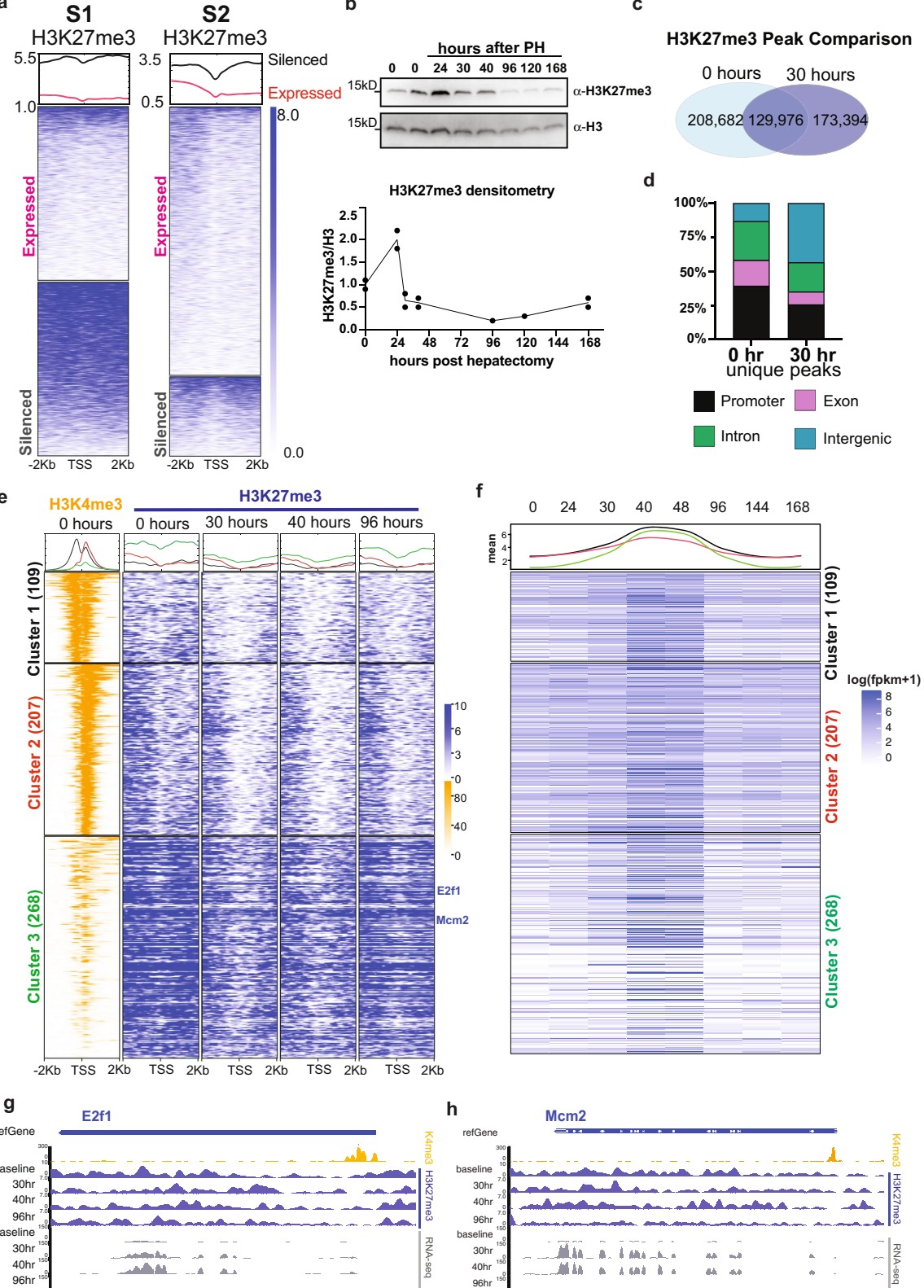

regeneration (Fig. 5f). GO analysis for 3 cluster further showed cell proliferation are specifically enrich in cluster 3 (Supplementary Fig. 6F). As an example, the important cell cycle regulator, *E2f1*, and *Mcm2*, a key component in DNA replication were included in Cluster 3 (Fig. 5e), with a high level of H3K27me3 and poised H3K4me3 occupancy at the TSS in quiescent livers, but with a depletion of H3K27me3 during regeneration

(Fig. 5g–h). These show that important cell cycle genes are silenced and maintained in a bivalent state in quiescent livers.

A very different pattern was found on liver enriched genes (Supplementary Dataset 2), which showed an elongated H3K4me3 signal around the TSS and no fluctuation of H3K27me3 during regeneration (Supplementary Fig. 6B). This confirms the finding that H3K4me3 domains spread more

**Fig. 5 H3K27me3 depletion primes cell proliferation genes for dynamic expression during liver regeneration. a** H3K27me3 occupancy around the TSS genes from S1 and S2 based on expressed and silenced categories demarked in Fig. 4B. **b** Western blot for H3K27me3 in liver samples collected at baseline and following PH. H3 is used as loading control. All the samples were processed in parallel. H3 and H3K27me3 were run in parallel on different gels. Each experiment was repeated independently with similar results and quantification of two biological replicates for all timepoints except for 96 and 120 h after PH is shown (lower panel). Dots represent biological replicates and line represent the trend of the change of H3K27me3 over time (Source data are provided as a Source Data file). **c** Comparison of H3K27me3 called peaks between baseline and 30 h after PH. **d** The peaks that were unique at each timepoint (i.e., from the nonoverlapping regions of the Venn diagram in **c**). were annotated by genomic elements to show that there is a redistribution from promoters to intergenic regions. **e** H3K4me3 and H3K27me3 occupancy around the TSS for genes residing in S1 and S2, which were categorized as cell proliferation related based on GO analysis. Genes were subject to K-means clustering based on H3K4me3 and H3K27me3 signals in quiescent livers. Metaplots retain the rank order of genes in each cluster across all the timepoints examined after PH. Gene number in each cluster is indicated. **f** Gene expression at 8 timepoints after PH is represented by log transformed FPKM + 1, with the rank order retained from panel **e**. Mean of expression at each timepoint for each cluster is plotted at the top and annotated by corresponding color for each cluster. **g** Epigenome browser view of *E2f1* and *Mcm2* expression and H3K27me3 occupancy during regeneration and H3K4me3 at baseline.

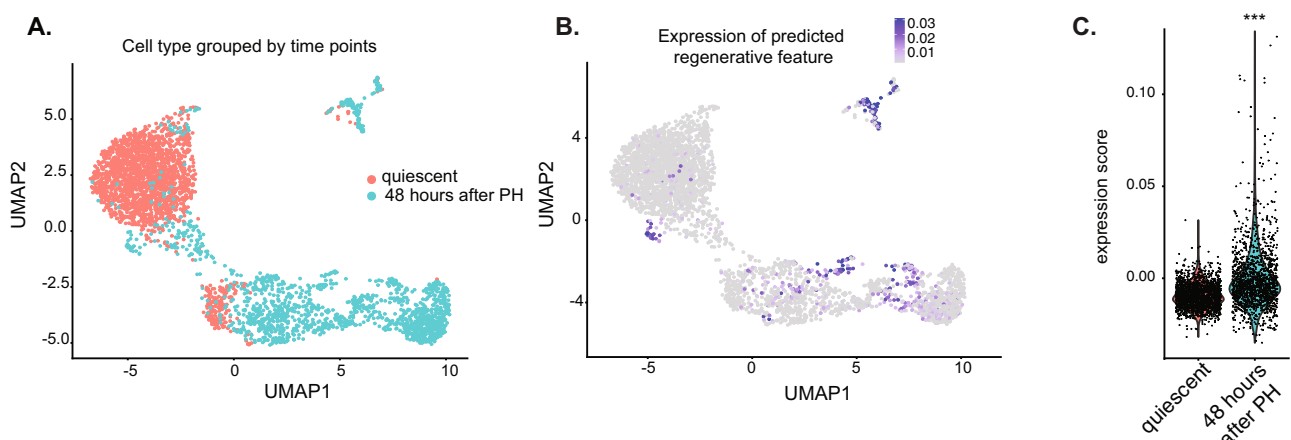

**Fig. 6 Proliferative genes are expressed in subset of regenerating hepatocytes. A** Uniform Manifold Approximation and Projection (UMAP) was used as a dimension reduction tool to visualize the complex connections among 4000 cells profiled from quiescent livers (orange) and 48 h (teal) after PH. **B** Expression level of the regenerative gene feature which summarizes the expression of all the genes analyzed in Fig. 5e into a single value is plotted across the same UMAP reduction as in **A**, highlighting only a few cells express these pro-regenerative genes at this timepoint. **C** Violin plot showing the expression score of the regenerative gene feature in each cell in quiescent livers and 48 h after PH. Welch *t*-test was performed on expression score between quiescent and 48 h, triple asterisks (***) indicate *p*-value < 2.2e-16 by two-side *t*-test.

broadly over genes that are essential for the identity and function of that specific cell type[64]. Most of these liver function genes were stably and highly expressed throughout regeneration (Supplementary Fig. 6C). Conversely, all the genes in S6 were devoid of H3K4me3, retained high levels of H3K27me3 and remained suppressed during regeneration (Supplementary Fig. 6D-E). The pattern of gene expression during regeneration was distinct from the pattern of TE expression, which remained largely stable over the time course of regeneration with SINEs and some LINEs being the highest expressed and DNA transposons being the lowest (Supplementary Fig. 7A-B). There was little change in the pattern of DNA methylation at 96 h after PH (Supplementary Fig. 7C) with no correlation between TE expression and DNA methylation changes at this timepoint (Supplementary Fig. 7D). Together, these data indicate that there are distinct epigenetic patterns in the quiescent liver that dictate the pattern of expression of genes essential for liver function, cell proliferation and for TEs during liver regeneration.

**Predicted regenerative genes feature a subset of hepatocyte during liver regeneration**. We next asked if all hepatocytes have the same transcriptional response during regeneration by mining a single-cell RNA-seq (scRNAseq) dataset from a mouse PH model[51] where the strain and sex of the samples match those used in our experiments. Analysis of nearly 4000 cells from quiescent livers and those collected at 48 h after PH grouped the cells into three distinct

clusters (Fig. 6A). The geneset identified in Fig. 5e was restricted to only a few cells in regenerating hepatocytes, and those with highest expression were distinct from the majority population (Fig. 6B), with some cells showing very high expression and others showing little to none (Fig. 6C). This reveals heterogeneity in the transcriptional response to the loss of liver mass. To determine if the same heterogeneous response occurs during regeneration in response to liver injury, we analyzed a second scRNAseq dataset generated from mice fed a diet that induced cholestatic liver disease[65]. This also revealed a heterogenous pattern of expression of the cell proliferation geneset in both hepatocytes and biliary cells in this model (Supplementary Fig. 8A-B), whereas the liver function genes were more heterogenous and largely restricted to hepatocytes. This indicates that these findings can be extended to other cell types and is relevant to liver disease.

**Discussion**

Epigenetic patterns are a well-established mechanism that coordinate gene expression, genome organization, and TE suppression; however, little is known about how epigenetic patterns contribute to gene expression in the liver or how they impact liver regeneration. The vast majority of studies on liver regeneration focus on how signaling pathways or specific transcription factors influence transcriptional programs. We investigated the epigenetic patterns that demarcate gene expression and transposon suppression in the liver under physiological conditions and

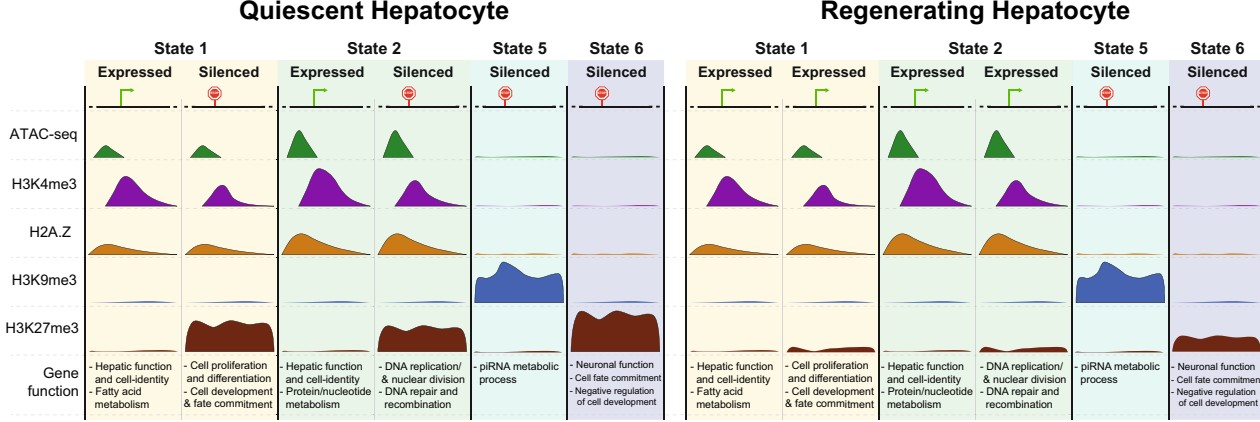

**Fig. 7 Model depicting the epigenetic combinatorial code change during liver regeneration.** Gene functions were annotated using GO analysis shown in Fig. 4.

asked whether an epigenetic code primes pro-regenerative genes to respond to acute loss of hepatic mass. This uncovered six distinct chromatin states in the mature male mouse liver corresponding to epigenetic marks associated with open chromatin (S1–S3), two of which are characterized by actively transcribed genes (S1–S2). We uncovered a landscape of constitutive heterochromatin marked by H3K9me3 and DNA methylation (S5), facultative heterochromatin marked by H3K27me3 (S6) and found that the vast majority of the genome (S4) is decorated by DNA methylation but largely devoid of the marks profiled by us and by ENCODE. H3K27me3 emerged as a key element of the epigenetic code that primes genes for differential expression during liver regeneration, but that TEs remain suppressed even when the epigenetic landscape is repatterned during regeneration. Together, these data support a model (Fig. 7) whereby an epigenetic code in quiescent mouse livers segments TEs into repressive states and positions the genes required for rapid response to stimuli—such as those that drive the restoration of liver mass—in open chromatin states but restrains their expression with H3K27me3 occupancy until the signal to regenerate is triggered. Surprisingly, we uncovered a heterogenous expression pattern of genes involved in cell cycle in regenerating livers in the context of chronic liver disease and PH. This suggests only some cells robustly express the regenerative geneset that we and others have studied. This confirms and extends the study by Chen et al.[51] who identified differences in gene expression in regenerating hepatocytes based on their location in the liver (i.e., the zonal expression pattern).

Other recent studies have illuminated the important role of epigenetic regulators in liver regeneration in the setting of chronic liver injury. One showed that differentiated biliary cells can exit a committed state in response to liver damage by remodeling the methylome/hydroxymethylome landscapes[66]. ATACseq profiling revealed that the chromatin became repatterned in regenerating livers so that pro-proliferation genes shifted to more open chromatin and the regulatory regions of genes involved in metabolism became more closed[18]. The interesting finding that the SWI/SNF complex can set a permissive chromatin state in hepatocytes to enhance liver regeneration by facilitating a gene expression profile characterizing liver-progenitor-like cells (LPLC)[17] is of particular interest. Our preliminary analysis points to commonalities between the open chromatin states identified by our work and the permissive states defined by Li et al.[17]: 77.19% of the LPLC genes reside in S1, S2, and S3, with the majority being encompassed by S2 (Supplementary Fig. 8D). This suggests that pre-existing accessible chromatin status defined by Li et al. is similar to the open chromatin states we identified.

It is of interest to define the elements of the epigenetic code that facilitate liver regeneration. We discovered that H3K27me3 was enriched on the promoters of a subset of genes that reside in active regions. Time-course transcriptomic analysis confirmed that genes that are bivalently marked with H3K27me3 and H3K4me3 and are also located in active chromatin states in quiescent livers are poised for differential regulation during regeneration. The expression patterns of such bivalent genes during regeneration mirrored the fluctuation of H3K27me3 occupancy on their promoters. This raises the question of how H3K27me3 is patterned on certain group of genes or genomic regions. The PRC regulates H3K27me3 in a variety of contexts[67] and work to elucidate DNA sequence determinants of PRC2 recruitment in mammals indicates that CpG islands are generally sufficient to drive H3K27me3 occupancy[68–70]. Our data point to a complex mechanism by which both H3K27me3 levels and genome occupancy change during liver regeneration, which could be due, in part, to changes in the expression of H3K27me3 modifiers, although the expression pattern of these genes does not fully explain the pattern of H3K27me3 levels or redistribution that we observed. It is possible that DNA methylation changes could induce redistribution of H3K27me3, as previously shown through studies where the hepatocyte genome was rendered hypomethylated by loss of Uhrf1[2]. However, analysis of DNA methylation at one timepoint following PH did not reveal substantial methylome changes, suggesting this as less likely. Other possibilities to be explored include the involvement of accessory proteins of PRC2 which fine tune H3K27me3 distribution[71], active demethylation, histone turn over, and delocalization from the chromatin.

The findings presented here provide a perspective on the packaging and potential suppressive mechanisms regulating TEs, which, if unleashed, can cause genome instability, change gene expression and induce an immune response[44,72,73]. In the mouse liver, over 33% of methylated cytosines reside in TEs[19,39], reflecting the essential role of DNA methylation in keeping TEs repressed. Our previous work uncovered striking cellular defects and gene expression changes caused by changes in DNA methylation[2,44,59,74,75], many of which we attribute to TE activation. Recent genome-wide profiling studies uncovered a complex pattern of repressive epigenetic marks on different TE families[47], and in vitro studies revealed a unique TE expression signature induced in cells with depleted different epigenetic regulators[47,48]. Here, we find that TEs are mainly occupied by DNA methylation and H3K9me3 in repressive chromatin states. By analyzing TEs based on age—with the older TEs being less dangerous since they degenerate and lose the ability to mobilize—we discovered H3K9me3 and DNA methylation are enriched as

together on young TEs, whereas H3K27me3 does not appear to coordinate with DNA methylation in TE occupancy. These findings implicate two of the best studied heterochromatin marks —H3K9me3 and CpG methylation—as key elements of the code that restrains the most dangerous TEs. This is supported by findings that blocking the writers of these modifications causes a similar subset of endogenous retroviruses (ERVs) and LINEs to be activated[48], demonstrating the functional importance of this repressive code.

Future work to explore how a combinatorial code dictates which genes are expressed and which are primed for expression when the liver is stimulated to regenerate will provide mechanisms to manipulate their functional impact. A recent study showing dramatic changes in the epigenetic landscape of the regenerating zebrafish fin report increased chromatin accessibility during regeneration and the static nature of DNA methylation during this process[3]. This finding is consistent with our data and with a study of chromatin accessibility in a model of liver regeneration in response to total hepatocyte ablation[18]. This suggests that mechanisms of regeneration are shared across species, even though the blastema model of fin regeneration is markedly different from liver regeneration. An important future goal is to uncover the transcriptomic and epigenomic features that dictate which cell subtypes repopulate the liver in response to injuries or resection.

## Methods

**Animal procedures**. Mice maintenance and all the experimental procedures were approved by the Institutional Animal Care and Use Committee (IUCUC) at either Icahn School of Medicine at Mount Sinai (07-0589) or NYU Abu Dhabi (17-0001A1). Temperature, humidity, and light/dark cycles were controlled and mice were fed food and water ad libitum. All the experiments were performed on male mice on a congenic C57Bl/6 background between 8 and 12 weeks of age. PH was carried out by removing 70% of the liver mass as described[76]. In brief, mice anaesthetized by isoflurane underwent a small incision in the abdomen, and two cuts to free the liver from the falciform ligament and the membrane that links the caudate and the left lateral lobe. A 4-0 silk thread (Ethicon, SA10) was placed at the base of the left lobe and tied prior to resecting the left lobe and another was tied around the median lobe right above the gall bladder, followed by resection to remove 2/3 of the median lobe. The peritoneal cavity was closed using 5-0 suture (Ethicon, JV389) and the skin was closed using wound clips (BrainTree Scientific, EZC APL)[2]. At 24, 30, 40, 48, 96, 120, and 168 h, and 4 weeks following surgery, mice were euthanized for liver collection, flash frozen in liquid nitrogen, and stored at −80 °C for subsequent analysis.

**RNA and DNA extraction**. RNA was isolated from liver tissues stored at −80 °C by first homogenizing using a Dounce homogenizer in DNA/RNA shield (provided by Zymo ZR-Duet kit), incubated with protease K (provided by kit) for 30 min at 55 °C, then column extracted using the Zymo ZR-Duet DNA/RNA MiniPrep kit following the manufacturer's instructions for silicon column-based RNA and gDNA extraction. Genomic DNA was extracted from liver samples using Qiagen DNase kit according to the manufacturer's instruction.

**Nuclear protein lysates and western blotting**. Male mice on a congenic C57Bl/6 background were used between 8 and 12 weeks of age to perform partial hepatectomy (PH). Samples from four quiescent livers were collected as the portion of the liver removed for PH. Regenerating livers were collected at 24, 30, 40, 96, 120, and 168 h after PH; 2 biological replicates were collected for all timepoints except for 96 and 120 h after PH. Livers were flash frozen in liquid nitrogen and stored at −80 °C; Nuclei were isolated from 20 mg of frozen tissue homogenized in 1.25 ml of buffer (5 mM MgCl2, 50 mM Tris-HCl pH 7.6, 50 mM NaCl, 1 mM EDTA, 5% v/v Glycerol, 0.1% v/v Triton X-100, 0.1% v/v b-mercaptoethanol) in a Dounce. Homogenates were centrifuged at 1100 × g for 10 min at 4 °C, resuspended in 0.5 ml of sonication buffer (150 mM NaCl, 150 mM Tris-HCl pH 7.4) and sonicated with a probe sonicator (Branson). Protein lysates were cleared by centrifuging at 11,000 × g for 15 min at 4 °C. The proteins in the supernatant were quantified using Qubit reagent (Invitrogen) and prepared by adding SDS-PAGE loading Leammli buffer (BioRad) and incubated at 95 °C for 5 min. Five micrograms of protein was loaded onto 4–20% precasted gels (BioRad), electrophoresed, transferred onto PVDF membranes (BioRad), blocked with 5% w/v powdered milk in TBST buffer (20 mM Tris-HCl, 150 mM NaCl, 0.1% v/v Tween 20, pH 8.0) for 1 h at room temperature, and incubated overnight at 4 °C with rabbit anti-H3 (Santa Cruz sc-10809-R, 1:5000) or monocolonal anti-H3K27me3 (Active Motif 61017-M, 1:1000), diluted in blocking buffer. After washing with TBST and incubation for 1 h with anti-Rabbit IgG HRP Conjugate (Promega, 1:2500) or anti-Mouse IgG, HRP

Conjugate (Promega, 1:2500) diluted in blocking buffer followed by washing in TBST, membranes were visualized using Pierce™ ECL Western Blotting Substrate (Thermo Fisher Scientific) or Clarity ECL substrate (BioRad) on the BioRad ChemiDoc. Immunoblot bands are quantified by densitometry using GelAnalyzer (http://www.gelanalyzer.com) and plotted using GraphPad Prism.

**ChIP sequencing**. Hundred to two hudredred milligrams of flash-frozen liver tissue was homogenized with a tissue Dounce as previously described and subsequent micrococcal digest, chromatin immunoprecipitation, and library preparation were carried out as described[2]. The antibodies used for ChIPseq and the concentrations, volumes and sources are anti-H3K4me3 (Abcam, ab1012 1 μg/μl; used 6 μl), anti-H3K27me3 (Active Motif, 61017; 1 μg/μl; used 6 μl), anti-H3K9me3 (Active Motif, 39161; 1 μg/μl; used 6 μl), and anti-H2AZ (Abcam, ab4174; 1 μg/μl; used 10 μl). The prepared libraries were sequenced on the Hiseq2500 platform for 75 bp single-end or 100 bp paired-end reads read runs at the Core Technology Platforms (CTP) at New York University Abu Dhabi (NYUAD). After sequencing, 75 base-pair single-end or 100 base-pair paired-end reads that passed quality trimming were aligned against the mouse reference genome (GRCm38.p4) using BWA-MEM[77]. The resulting BAM files were then processed through PICARD tools (to clean, deduplicate and sort) for downstream analysis and visualization. One sample from each timepoint and antibody was used.

**ATACseq**. Hundred milligrams of liver tissue was homogenized with a tissue Dounce and 50,000 nuclei were used to prepare libraries following the ATACseq protocol of Buenrostro et al.[28]. Libraries with the expected size distribution determined by bioanalyzer were sequenced on the Hiseq2500 platform for 100 cycles single-end read runs at the Genomics Core facility of the NYUAD. Sequenced reads were aligned to the mouse reference genome (GRCm38.p4) with BWA-MEM, generating sorted and cleaned bam with PICARD for further analysis. One liver was used.

**Gene expression profiling**. One microgram of total RNA was used to generate libraries. RNA-seq libraries were prepared with poly-A capture according to the Illumina TruSeq RNA sample preparation version 2 protocol, following manufacturer's instruction. The prepared libraries were sequenced on the Hiseq2500 platform for 100 cycles, paired-end read runs by the NYUAD Bioinformatic Core. Sequencing quality was assessed using FastQC and the raw reads were quality trimmed using Trimmomatic[78] to remove low Q scores, adapter contamination and systematic sequencing errors. The reads that passed quality control were aligned to reference genome GRCm38.p4 by using tophat2 v2.1.0, with the parameters "–no-novel-junctions" and "–G"[79]. Overall alignment rates were above 95%. Three biological replicates were adopted for all experiments; replicates that were identified as outliers by principal component analysis were excluded and in those cases two samples were used.

**Single-cell RNA-seq analysis**. Two scRNAseq datasets generated from regenerating mouse livers[51,65] were retrieved from GSE158874 and GSE125688. The standard 10X Genomics Cell Ranger output was downloaded and imported using the Read10X function in Seurat (version 3.0)[80]. Hepatocytes isolated from quiescent (i.e., baseline) livers were downsampled 2000 cells to match the cell numbers from livers collected at 48 h after PH. A Seurat object was created by merging and labeled the cells from both timepoints. The quality control step in Seurat to retain cells with unique gene counts that were between 200 and 6000 and the percentage of mitochondrial genes that was lower than 75%. After removing the unwanted cells, the data were normalized using the "LogNormalize" method with a scale factor of 10,000 and scaled to remove the unwanted sources of variation from mitochondrial gene content and number of detected unique molecules in each cell. The top 2000 variable genes were calculated using the FindVariableFeatures function in Seurat and used in the following principal component analysis. Uniform Manifold Approximation and Projection (UMAP) was used as the dimension reduction. Gene feature expression was calculated with the function AddModuleScore and plotted in the same UMAP reduction. In the liver epithelium response to injury dataset, the data matrix containing contain number of unique UMI-corrected transcripts per gene per cell for BEC_ctr1, BEC_DDC1, HEP_ctr1, and HEP_DDC1 are downloaded and imported in Seurat. Four matrixes are integrated with the function of FindIntegrationAnchors in Seurat. The QC, clustering, scaling and PCA are referred as the first dataset. Dimension reduction is performed with UMAP so that the regenerative gene feature and liver function gene feature are plotted with the module score.

**DNA methylation profiling**. eRRBS was previously performed on genomic DNA from mouse livers at baseline and 96 h after PH[39]. Bismark[81] was used for alignment and calling CpG methylation with default parameters. Summary and visualization of CpG methylation was with R package 'methylKit'[82]. CpGs with methylation level below 20% were treated as unmethylated and above 80% as methylated. Since transposons are of unequal length and not always highly conserved in sequence, we created "metaplots" for transposons and nontransposons dividing each sequence into 40 equal bins for analysis and plotted winsorized mean values (1–99 percentile) for each bin. Two biological replicates were analyzed.

**Data analysis and visualization**. Chromatin states were predicted from aligned binarized bam files above by using the LearnModel command in ChromHMM[34], where the number of states was set as 6 after optimization. To quantify gene expression, Cufflinks v2.2.1 was used to derive FPKM values, while read counts were generated using HTseq[83] count. In ChIPseq, peak calling was with Macs2 with parameters adjusted according to different markers, H3K4me3 and H3K9me3 in narrow peak mode, H3K27me3 in broad peak mode and ATACseqw in broad peak with -f BAMPE, p-values are all set 0.05. Heatmaps of gene expression and ChIPseq signals were generated with R package named EnrichedHeatmap with normalization of genomic signals within target regions[84]. All targeted regions were customized within an annotation package from Bioconductor (https://www. bioconductor.org/). *TxDb.Mmusculus.UCSC.mm10.knownGene: Annotation package for TxDb object(s)* in R package version 3.4.7. To plot ChIPseq signals with mapped reads, we used deepTools[85] generating bigwig files, normalized with RPKM for genome browser view. Clustering of heatmap and profile plot in baseline of H3K4me3 and H3K27me3 was set K-means as 3. The epigenome browser was used to generate all browser view figures[86]. Gene Ontology analysis was implemented with the R package ClusterProfiler, adopting the biological process class of GO and setting p-value as 0.05. Box and density plots for methylation analysis were performed with ggplot2 R package. Metaplot for CpG density was performed on *mm10* using the genomation R package.

**Statistical analysis**. Test of differential expression uses a generalized linear model. We took the HTseq[83] counted reads of genes at each timepoints compared them to baseline, respectively. The gene reads were modeled as negative binomial distributions. For multiple testing, we apply Benjamin–Hochberg correction, which was also implemented in DEseq2 in Bioconductor, with adjusted p-values < 0.05 as statistically significant. For transposable elements quantification, by taking the same alignment for gene quantification, reads were counted by HTseq[83] with union mode and not strand-specifically according to a RepeatMasker annotation of mm10 from UCSC table browser as the reference. The TEs quantification is family based. Statistical analysis is implemented with DESeq2 using the same parameters as employed for differential gene expression analysis, with simple repeats excluded.

To accommodate the large number of TE copies and high variance in the sw Score, we sampled 1000 copies of TEs in each group, then permuted sampling 100 times to generate a normalized distribution representing the age of categorized TEs. We then adopted t-test to test the significance of TE ages in each category compared to the age of all TEs. To compare gene expression between those with H3K27me3 or without, we downsampled genes without H3K27me3 to the same number as the group with H3K27me3, and then plotted the log transformed FPKM. Statistical analysis on CpG methylation levels were performed with Prism Graphpad 8 using ANOVA or t-test.

**Reporting summary**. Further information on research design is available in the Nature Research Reporting Summary linked to this article.

## Data availability

Transcriptomic profiling of WT mouse liver at baseline and following PH are at "GSE125007". Genome-wide profiling for H3k4me3, H3k27me3, and H3k9me3 in WT mouse liver are at "GSE125006". Genome-wide profiling for H2A.Z and genome accessibility (ATACseq) in in WT mouse liver are at "GSE153090". Genome binding/ occupancy profiling in mouse liver by high throughput sequencing from ENCODE "GSE31039". Source data are provided in published manuscripts: Chen et al. 2020 and Pepe-Mooney et al. 2019: Single-cell omics analysis of hepatocytes during liver regeneration is from "GSE158874"[51] and Single-Cell Analysis of the Liver Epithelium in Homeostasis and Regeneration is from "GSE125688"[65]. All other relevant data supporting the key findings of this study are available within the article and its Supplementary Information files or from the corresponding author upon reasonable request. A reporting summary for this Article is available as a Supplementary Information file. Source data are provided with this paper.

## Code availability

R code for all downsteam analysis and visualization is available at: https://github.com/ zcmit/Liver-Regeneration/releases/tag/v1.0.0 https://doi.org/10.5281/zenodo.4718103.

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

## Acknowledgements

We are grateful for input from the NYUAD Center for Genomics and Systems Biology, especially S. Wang for initial study and data generation, N. Drou for sequencing data processing insight, Xiyuan Zhang for insights on PRC2, and scRNAseqq analysis. B. Madakashira for preparing ATACseq libraries, S. Ranjan and B. Madakashira for the expert animal maintenance and members of the Sadler lab for critical discussion of the manuscript. This study was funded by the NYUAD Faculty Research Fund and the National Institute of Diabetes, Digestive and Kidney Diseases (2R01DK080789) to K.C.S.

## Author contributions

C.Z. and K.C.S. conceived of the project; C.Z., K.C.S., and F.M. designed experiments, C.Z. carried out the data analysis and F.M. carried out the RRBS analysis. F.M. and E.M. carried out the experiments; C.Z., F.M., and K.C.S. contributed to results interpretation and wrote the manuscript. All authors read and approved the final manuscript.

## Competing interests

The authors declare no competing interests.
