## [Peer Review File · Nature Communications]

REVIEWER COMMENTS

Reviewer #1 (Remarks to the Author):

The manuscript presented by Zhang C et al describes in a very precise way how the epigenetic code in quiescent liver tissue dictates the patterns of gene expression required in the liver regenerative process after a partial hepatectomy. Through a combinational analysis from the ENCODE datasets together with Chip-seq, ATAC-seq, DNA methylation and gene expression profiling they define six distinct chromatin stages, which will be classified by open chromatin, characterized by actively transcribed genes or constitutive and facultative heterochromatin. The methylation pattern in each one of them, its association with the studied histone marks, its relationship with Transposable Elements and the detailed description of localization and identification in the different states is very well described. A very interesting aspect found is the fact of finding H3K27me3 is enriched on the promoters of active regions, for example in cell cycle regulatory genes poised for expression in quiescent livers and expressed during regeneration. The work is perfectly designed and carried out, with all rigorously established analysis. Furthermore, the results provided in this manuscript lay the foundations to numerous and highly applicable studies. In this sense, the premises of the present work establish how knowledge of the epigenetic landscape can further predict the regenerative potential capacity. Several issues related with this potential applicability arise,

- The first results show how investigating ENCODE dataset profiling on available Chip-seqs where the six states were identified. On the other hand, the study reveals that more epigenetic marks would add very valuable information, how feasible would it be to integrate ENCODE analysis with marks such as H3K27ac, H3K4me1, H3K79me3 with the data reported in the paper?
- To be able to predict the regenerative potential capacity, it is important to know how the epigenetic landscape will be orchestrated in situations where the regenerative capabilities are compromised. Given that it has been shown that the regenerative capacity of the liver decreases markedly with age, it would be very interesting to see to what extent the liver of old mice present differences in the entire epigenetic panorama (changes in the different chromatin stages, in histone marks associated, TE or DNA methylation) to establish if specific changes could be responsible of this effect.
- It has been shown the dynamic and highly complex role for H3K27me3 in gene regulation and genome organization in the liver, and how specific depletion of H3K27me3 occurs on key genes that are highly regulated during liver regeneration. The dynamic expression that writers and readers present during the regenerative time points do not explain properly the fluctuation of H3K27me3 mark. Which other explanations could contribute to explain all these phenomena?
- Given the extensive and detailed analysis of DNA methylation in specific CpG and Transposable Elements, it would be very interesting to know in detail if these TE would experience changes during the regenerative process, as well as the specific DNA methylation in each time point of the process.

Reviewer #2 (Remarks to the Author):

The authors investigate the hypothesis that liver regeneration is "encoded by the epigenome" of quiescent hepatocytes. They proceed by utilizing ChIP-seq data for several histone modifications (and H2A.Z) as well as ATAC-seq as input to ChromHMM to define chromatin 'states' of the quiescent hepatocytes. Integrating these results with DNA methylation data they had previously generated, they characterize the types of regions that have particular states.

General Comments:

One major concern with the manuscript is that it is mostly confirmation of what we already know

about the patterns of chromatin modifications at regulatory elements, e.g. promoters in the mouse genome are mostly CpG islands that are generally not methylated and are marked by "active" histone modifications.

One of the more interesting observations reported is that genes involved in cell cycle have "active" states, but also H3K27me3 in quiescent livers, and that they lose H3K27me3 during liver regeneration. It would have been nice to see more work related to this.

Another concern I have with the paper is with the use of language describing chromatin marks as 'causal' for gene expression changes. For example, the authors state that waves of gene activation and repression during regeneration "implicates epigenetics as an underlying regulatory mechanism". Given that the enzymes responsible for depositing these marks have no obvious sequence specificity, it's hard to imagine them "regulating" genes in the same way as, e.g. transcription factors. Indeed, the "active" mark H3K4me3 that the authors consider is really just an indication that Pol II has already been recruited to a locus.

My last general comment is that the manuscript would benefit greatly from editing/proofing, there are sections that are hard to follow and some referenced figure panels (Figs 2 and 6 in particular) that don't align between the text and the figures.

Specific Comments:

1. In Fig 2, the authors show that state 6 regions with H3K27me3 have an interesting DNA methylation pattern, with ~50% of the CpGs being methylated. In Fig 3, they show that H3K27me3 domains are mostly in either gene bodies or intergenic regions. This seems likely to explain the distribution in Fig 2 with the intergenic H3K27me3 domains mostly having DNA methylation. Can the authors comment on this?
2. The reference to Fig 2E in the text states that "Nearly all CpGs in regions covered by H3K9me3, regardless of the state, were fully methylated (Fig. 2E)." However, Fig 2E does not appear to show much DNA methylation at all.
3. In Fig 3 the authors define "promoters" as the 500bp upstream of the TSS. This is very small for the mouse genome. Is there a reason the authors are using such small windows as promoters? This will lead to many "promoter" type regions being considered as intergenic.
4. In motivating Fig 4, the authors argue that the "simple model" of TE repression is that they are packaged into heterochromatin with DNA methylation as the "key repressive mark". This is really only true of evolutionarily young TEs. As TEs age, they mutate and lose the ability to jump / function in the genome and are no longer required to be silenced. I think the results shown here are supportive of this model so it would be nice to see a discussion of that.

Reviewer #3 (Remarks to the Author):

This interesting study profiled the epigenetic landscape in quiescent hepatocytes and the regenerating liver after partial hepatectomy. The authors identified six chromatin states and provided substantial epigenetic resources of mouse liver. Interestingly, the authors found pro-regenerative genes, such as cell cycle genes, are bivalent which are maintained in open chromatin states but restrained by H3K27me3. During liver regeneration, these genes expressed and lost H3K27me3 modification, suggesting an epigenetic mechanism for a quick regenerating response.

- 1, The authors concluded that bivalent domains occupied by H3K27me3 and H3K4me3 in quiescent livers silence cell cycle genes, poising them activation upon mitotic stimulation to allow for liver regeneration. However, Figure 6A showed that H3K4me3 and H3K27me3 seemed to be reciprocal around TSS of cell cycle genes. The author should perform clustering on these genes according to H3K4me3 and H3K27me3 and show how many (percentage) genes were indeed bivalent.
- 2, Given that the bivalent marked cell cycle genes are activated during regeneration with reduction of H3K27me3, it is interesting to learn bivalent marked genes which are not related to cell cycle but to other hepatic regeneration. In addition, how many pro-regenerative genes are bivalent?
- 3, The authors identified cell cycle genes were bivalent, which were pre-opened and marked with H3K27me3 and H3K4me3. Are these chromatin state and modifications resolved after PH? How about other chromatin features change at these genes during regeneration after PH?
- 4, The authors suggested that bivalent feature delivered marked genes a potential for regeneration. Permissive state was found in quiescent hepatocytes, which contributed to liver regeneration. For example, the study by Li et al (2019, Cell Stem Cell 25, 54-68) has already described a pre-existing accessible chromatin status in quiescent livers, which should be thoroughly discussed or compared in this paper.
- 5, Transcriptome after PH was analyzed in detail by Wang et al. (2019, Developmental Cell 50, 43–56). The distinct expression kinetics of genes may partially reflect their chromatin states. Could the authors discuss or perform additional characterizations of the chromatin state of these genes?
- 6, An interesting finding is that the combination of H3K9me3 and DNA methylation marked the youngest transposons. Are there any special features of these TEs related to liver biology?
- 7, The authors classified chromatin status by chromHMM. However, there were some inconsistency in the study. 1) the bivalent chromatin which was highlighted in this work was not found in six chromatin status in Figure 1A; 2) in Figure 5, ATAC, H3K4me3, H2AZ and H3K27me3 were detectable in S4, which was proposed as a low signal chromatin. Authors should provide analysis to explain these inconsistencies and showed a quality control of ChIP-seq data.

Minor concerns

- 1, At Page 10, there is no Fig2F. Fig2E and Fig2F probably mean Fig2D and Fig2E.
- 2, It was read that "As expected, while all genes in S5 were covered with H3K27me3, very few also show occupancy by H3K4me3". In Fig. 5a, no H3K27me3 enrichment in S5 was found.
- 3, In Fig3, the enrichment of genomic elements in ChromHMM states needs statistical analyses.
- 4, Fig5C implies that liver enriched genes almost absent from S3, S5 and S6. However, this figure highlights the orange points as liver enriched genes, which seem to be equally distributed in several classes. The authors may provide the proportion of liver enriched genes in S3, S5 and S6.

Reviewer #1 (Remarks to the Author):

The manuscript presented by Zhang C et al describes in a very precise way how the epigenetic code in quiescent liver tissue dictates the patterns of gene expression required in the liver regenerative process after a partial hepatectomy. Through a combinational analysis from the ENCODE datasets together with Chip-seq, ATAC-seq, DNA methylation and gene expression profiling they define six distinct chromatin stages, which will be classified by open chromatin, characterized by actively transcribed genes or constitutive and facultative heterochromatin. The methylation pattern in each one of them, its association with the studied histone marks, its relationship with Transposable Elements. and the detailed description of localization and identification in the different states is very well described. A very interesting aspect found is the fact of finding H3K27me3 is enriched on the promoters of active regions, for example in cell cycle regulatory genes poised for expression in quiescent livers and expressed during regeneration. The work is perfectly designed and carried out, with all rigorously established analysis. Furthermore, the results provided in this manuscript lay the foundations to numerous and highly applicable studies. In this sense, the premises of the present work establish how knowledge of the epigenetic landscape can further predict the regenerative potential capacity. Several issues related with this potential applicability arise,

We appreciate the reviewer's positive comments and are grateful for the insights provided to improve the manuscript. We have addressed each of these comments and have generated new data and extended the analysis in addition to updated text. We believe the revised version of the manuscript is improved.

1. The first results show how investigating ENCODE dataset profiling on available Chip-seqs where the six states were identified. On the other hand, the study reveals that more epigenetic marks would add very valuable information, how feasible would it be to integrate ENCODE analysis with marks such as H3K27ac, H3K4me1, H3K79me3 with the data reported in the paper?

We are very interested in using publicly available datasets to enhance our work, and have revisited the ENCODE data as this reviewer suggested. In the original manuscript, we mined all the available ChIP-seq data generated for epigenetic regulators from adult mouse liver samples which were comparable to the samples for the data we generated (i.e. same strain, sex and age) and tried to integrate them. In theory, this could extend the

coverage of the genome, as the ENCODE datasets include epigenetic marks that we did not profile. However, we found a batch effect in part due to greater sequencing depth and library enrichment in the samples we generated for H3K4me3 and H3K27me3, meaning that even if the mark was the same, the profiles were dissimilar enough from the ENCODE datasets that they could not be combined without risking attenuating the performance of the algorithm we adopted for this study.

*To investigate whether this limitation resulted in a significant loss of information (i.e. was there a substantially greater coverage of the genome by the marks profiled in ENCODE), we compared the regions that fall into the 'empty' states based on the chromatin states determined using our datasets (state 4; S4) and the ENCODE dataset (E-S5). We show in **NEW Figure S2D**, that the 'empty' states almost entirely overlap, with only 21.5% of the regions in S4 and 7.2% of the regions in E-S5 as unique to each dataset. This is consistent with the conclusion that regulatory elements, which are the primary focus of the marks profiled in ENCODE, account for a small fraction of the mouse genome (Weitzman 2002).*

2. To be able to predict the regenerative potential capacity, it is important to know how the epigenetic landscape will be orchestrated in situations where the regenerative capabilities are compromised. Given that it has been shown that the regenerative capacity of the liver decreases markedly with age, it would be very interesting to see to what extent the liver of old mice present differences in the entire epigenetic panorama (changes in the different chromatin stages, in histone marks associated, TE or DNA methylation) to establish if specific changes could be responsible of this effect.

This interesting question outlines a part of our future research plans to identify how chromatin states change with aging. We generated new epigenomic and transcriptomic data from aged mice and have just started to analyze this in detail. We share the progress here for the reviewer's benefit but there is more to do, and therefore have not included this data in the manuscript.

To determine if aging alters the hepatic chromatin state and if this altered the gene expression profile, we selected those epigenetic features that appeared to be the most informative for regulating pro-regenerative genes (ATAC-seq and ChIPseq for H3K4me3 and H3K27me3) and profiled them in the liver of 14 month old male mice. We then compared the occupancy across the cell cycle with the patterns in young mice (Reviewer Figure1 A). We compared the occupancy across the pro-regenerative genes from S1 and

S2 as well as the liver enriched genes between young and 14 months old male mice and did not observe a significant difference (Reviewer Figure 1A-B). We detected weaker peak signals in aged mice, and we are currently working to determine if this reflects a biological difference or could reflect different sequencing depth of the young and aged samples since they were run separately. The time scale and resources available to respond to the reviewer comments has limited the amount of analysis that we could do on these samples and therefore we plan to expand this for a future study.

Since the effects of C/EBP α on regenerative capacity of aging livers has been shown to be mediated by repression of E2F family members (Iakova, Awad, and Timchenko 2003), we took a different approach to examine the effects of aging on the hepatic chromatin state by investigating the chromatin environment on E2F1. We found that E2F1 is bivalent in quiescent livers and is found in cluster 3 (NEW Figure 5E). This suggests the interesting possibility that chromatin state changes in aged mice could shift this gene into a more repressive state, perhaps by depleting H3K4me3 or addition of another repressive mark.

In addition to the new ChIPseq and ATACseq data, we also performed RNAseq analysis on these same livers from 14 month old mice. We found minimal changes in gene expression compared to young mice (Reviewer Figure 2).

Interestingly, the transcripts that were differentially expressed were mostly pseudogenes.

This is consistent with recent findings from others showing that pseudogene derepression reflects changes in genome remodeling and is attributed to retrotransposition of transposable elements (Sisu et al. 2020). This suggests that there are epigenetic changes between young and old mice and a more detailed analysis along with additional datasets are required to address this. However, the scope and time frame for these studies far exceeds the time frame for revising the current manuscript.

3. It has been shown the dynamic and highly complex role for H3K27me3 in gene regulation and genome organization in the liver, and how specific depletion of H3K27me3 occurs on key genes that are highly regulated during liver regeneration. The dynamic expression that writers and readers present during the regenerative time points do not explain properly the fluctuation of H3K27me3 mark. Which other explanations could contribute to explain all these phenomena?

*We are very interested in gaining insight into this very interesting question. In the revised manuscript, we provide additional data that adds to the understanding of how H3K27me3 may be regulated during regeneration. To extend the expression analysis based on RNAseq, we assessed the total levels of H3K27me3 in the liver during regeneration using Western Blotting which shows a dynamic pattern of total levels of H3K27me3 (**NEW Figure 5B**). This pattern is largely consistent with findings we report for H3K27me3 ChIPseq, with minor discrepancies which could be attributed to using two very different experimental techniques. Data that was previously presented in a supplemental figure showed a redistribution of H3K27me3 at 30 hours after PH away from promoters and toward intergenic regions. We have now moved that graph to the main figure (**NEW Figure 5C-D**). Together, this points to a complex mechanism by which H3K27me3 levels change during regeneration, including the possibilities of demethylation, histone turn over and delocalization from the chromatin. Additionally, several accessory proteins of PRC2 were recently reported to be responsible for the proper distribution of H3K27me3 (van Mierlo et al. 2019). We discussed this important point in the revised manuscript.*

4. Given the extensive and detailed analysis of DNA methylation in specific CpG and Transposable Elements, it would be very interesting to know in detail if these TE would experience changes during the regenerative process, as well as the specific DNA methylation in each time point of the process.

*We have further explored this through new analysis and included a new dataset profiling DNA methylation at 96 hours after PH (**NEW Figure S7**). We quantified TE expression from RNAseq data at 6 time points after PH and then segregated the TEs based on class (**NEW Figure S7A-B**). This shows very minimal changes in expression of TEs during*

regeneration. We next compared the differences in DNA methylation and its effect on TEs expression at baseline and at 96 hours after PH using eRRBS (**NEW Figure S7 C-D**). This shows that 4.5% of CpG are hypo-methylated while only 1.83% of CpG are hyper-methylated at 96 hours (**NEW Figure S7C**). Integrating these methylation differences with TE expression changes showed no correlation and only negligible changes (+/- 5%) to the methylation of CpGs in TEs (**NEW Figure S7D**). This indicates that in normal liver regeneration, there is a very small impact, if any, on TE expression and little change in the DNA methylation pattern.

Reviewer #2 (Remarks to the Author):

The authors investigate the hypothesis that liver regeneration is “encoded by the epigenome” of quiescent hepatocytes. They proceed by utilizing ChIP-seq data for several histone modifications (and H2A.Z) as well as ATAC-seq as input to ChromHMM to define chromatin ‘states’ of the quiescent hepatocytes. Integrating these results with DNA methylation data they had previously generated, they characterize the types of regions that have particular states.

We appreciate that this reviewer provided important insights for refining our analysis and improving this manuscript.

1. One major concern with the manuscript is that it is mostly confirmation of what we already know about the patterns of chromatin modifications at regulatory elements, e.g. promoters in the mouse genome are mostly CpG islands that are generally not methylated and are marked by “active” histone modifications.

We appreciate that some of the big-picture findings of this study are consistent with epigenetic patterns found in other tissues. However, we argue that such findings do not negate the significance of this work. The findings reported here represent the first robust map of the hepatic epigenetic landscape in quiescent and regenerating livers. Moreover, our study proposes several highly novel concepts relevant to liver biology and regeneration. Further, the focus on the epigenetic pre-pattern in the liver that confers regenerative potential is new.

We found that H3K27me3 fluctuates during PH, specifically in a pattern that mirrors the expression of genes involved in cell proliferation. This therefore suggests that H3K27me3 as a key player in coordinating gene expression and highlights an important and novel

role of epigenetic regulation via H3K27me3 in liver regeneration. These findings are integrated with the emerging body of literature that shows that bivalent chromatin in the liver may be a general mechanism involved in liver processes. The finding that PRC2 proteins EZH1 and EZH2 regulate genes that control hepatocyte maturation and fibrogenesis by acting at euchromatic promoter regions (Grindheim et al. 2019), support our conclusions.

2. One of the more interested observations reported is that genes involved in cell cycle have “active” states, but also H3K27me3 in quiescent livers, and that they lose H3K27me3 during liver regeneration. It would have been nice to see more work related to this. (Genes that lost H3K27me3)

*We are pleased that the reviewer asked for more details regarding the pattern of H3K27me3 during liver regeneration. We performed a new analysis that grouped the cell cycle genes that were marked by H3K27me3 and resided in ‘active’ states into three clusters based on H3K4me3 occupancy (**NEW Figure 5E**). This identified three clusters of genes: one cluster (109 genes) was highly enriched with H3K4me3 arounds the TSS with relatively lower H3K27me3 occupancy, a second cluster (207 genes) was enriched with H3K4me3 in the gene body and a third cluster (268 genes) had lower H3K4me3 occupancy but had relatively higher H3K27me3 occupancy. The Cluster 3 genes are the most repressed in the quiescent liver and are among the most active during regeneration (**NEW Figure 5F**).*

*To further investigate the pattern of expression of the genes that lost H3K27me3 during regeneration, we asked if all or only a subset of hepatocytes express this regenerative response. We mined a set of single-cell RNAseq (scRNA-seq) in a mouse PH model (Chen et al. 2020), where the strain and sex of the samples match those used in our study. We examined the expression of the cell cycle genes that resided in the active state but were suppressed in quiescent hepatocytes in this dataset. Surprisingly, only few cells highly expressed this geneset, revealing heterogeneity in the hepatocyte response to the loss of liver mass (**New Figure 6**). To test the applicability of this dataset to the regeneration that occurs in response to liver injury, we analyzed a second scRNA-seq dataset generated from mice fed a diet that induced cholestatic liver disease (Pepe-Mooney et al. 2019). Interestingly, we found our geneset of interest was not only expressed in injured hepatocytes but also in the biliary cells (**NEW Figure S8A-C**). This suggests that our findings can be extended to other cell types and is relevant to liver disease.*

3. Another concern I have with the paper is with the use of language describing chromatin marks as 'causal' for gene expression changes. For example, the authors state that waves of gene activation and repression during regeneration "implicates epigenetics as an underlying regulatory mechanism". Given that the enzymes responsible for depositing these marks have no obvious sequence specificity, it's hard to imagine them "regulating" genes in the same way as, e.g. transcription factors. Indeed, the "active" mark H3K4me3 that the authors consider is really just an indication that Pol II has already been recruited to a locus.

We have reflected the comments from the reviewer in the revised text of the manuscript to indicate that the finding of histone marks on genes that are not directly regulatory and have extended the discussion about H3K27me3 in the edited manuscript.

4. My last general comment is that the manuscript would benefit greatly from editing/proofing, there are sections that are hard to follow and some referenced figure panels (Figs 2 and 6 in particular) that don't align between the text and the figures.

We apologize for the mistake in the figure annotations. This has been corrected.

5. In Fig 2, the authors show that state 6 regions with H3K27me3 have an interesting DNA methylation pattern, with ~50% of the CpGs being methylated. In Fig 3, they show that H3K27me3 domains are mostly in either gene bodies or intergenic regions. This seems likely to explain the distribution in Fig 2 with the intergenic H3K27me3 domains mostly having DNA methylation. Can the authors comment on this?

*We appreciate this comment; the point is correct. We attribute this to the binding features of H3K27me3; the peak of H3K27me3 is broad (Figure 1D), and covers both repressive promoters and bivalent promoters; there are also more peaks in the intergenic regions than those in the 'active' state. In comparison to the DNA methylation pattern, we found that CpGs in intergenic regions have a high probability of being methylated in contrast to promoters, where there is virtually no methylation. Thus, based on probability, the intergenic regions marked with H3K27me3 are likely to be methylated whereas the promoters marked by H3K27me3 are likely to be unmethylated. We have adjusted the presentation of our data to display the methylation levels in each state in a way that is clearer to understand (**NEW Figure 2B**). This pattern is consistent with studies that show that the presence of significant cytosine DNA methylation render CpG islands refractory*

to PRC2 binding (Lynch et al. 2012; Reddington et al. 2013). We conclude that H3K27me3 is deposited at CpG rich regions that fall into two categories: S1 and S2 around promoters which lack CpG methylation and S6, where it is largely in the intergenic region, where CpG methylation and CpG density are relatively high. We added explanation in the text to reflect this.

2. The reference to Fig 2E in the text states that “Nearly all CpGs in regions covered by H3K9me3, regardless of the state, were fully methylated (Fig. 2E).” However, Fig 2E does not appear to show much DNA methylation at all.

We have corrected this error.

3. In Fig 3 the authors define “promoters” as the 500bp upstream of the TSS. This is very small for the mouse genome. Is there a reason the authors are using such small windows as promoters? This will lead to many “promoter” type regions being considered as intergenic.

*We are grateful for this helpful comment. In our initial analysis, we used a range of promoter sizes (500 bp, 1000 bp and 2000 bp), and found that the 500 bp window was the most robust, and have updated the text to include the caveat that this may lead to the inclusion of promoter regions in areas we categorize as intergenic. Indeed, these are larger than the promoters that were characterized by others (Bajic et al. 2006). In subsequent analysis of the TSS occupancy of histone marks, we use a broader window (+/- 2,000 bp; Figure 5A, **NEW Figure 5E**).*

4. In motivating Fig 4, the authors argue that the “simple model” of TE repression is that they are packaged into heterochromatin with DNA methylation as the “key repressive mark”. This is really only true of evolutionarily young TEs. As TEs age, they mutate and lose the ability to jump / function in the genome and are no longer required to be silenced. I think the results shown here are supportive of this model so it would be nice to see a discussion of that.

We have expanded the discussion to include this observation.

Reviewer #3 (Remarks to the Author):

This interesting study profiled the epigenetic landscape in quiescent hepatocytes and

the regenerating liver after partial hepatectomy. The authors identified six chromatin states and provided substantial epigenetic resources of mouse liver. Interestingly, the authors found pro-regenerative genes, such as cell cycle genes, are bivalent which are maintained in open chromatin states but restrained by H3K27me3. During liver regeneration, these genes expressed and lost H3K27me3 modification, suggesting an epigenetic mechanism for a quick regenerating response.

We are grateful for the positive comments of this reviewer and have addressed all of the comments, which we believe has improved the manuscript.

1. The authors concluded that bivalent domains occupied by H3K27me3 and H3K4me3 in quiescent livers silence cell cycle genes, poising them activation upon mitotic stimulation to allow for liver regeneration. However, Figure 6A showed that H3K4me3 and H3K27me3 seemed to be reciprocal around TSS of cell cycle genes. The author should perform clustering on these genes according to H3K4me3 and H3K27me3 and show how many (percentage) genes were indeed bivalent.

*This is an excellent suggestion which we implemented in **NEW Figure 5E and 5F**. We performed K-means clustering based on H3K4me3 and H3K27me3 signals at baseline, categorized them into 3 clusters and then rank ordered based on H3K4me3 signal intensity and plotted the H3K27me3 occupancy at 30, 40 and 96 hours after PH. This significantly improve our bivalency profiling, and highlighted Cluster 3 as the most enriched in H3K27me3 and with the lowest H3K4me3 compared to the other clusters. Cluster 3 genes had the lowest expression level at baseline, and a higher level of activation during regeneration. We extended this by analyzing the liver function genes which are highly expressed in the liver and are marked by H3K4me3+, H3K27me3-. In **NEW Figure S6B-C**, we show that these genes have virtually no change in H3K27me3 and their expression is stable during regeneration. This not only serves as an internal control for the findings presented in Figure 5, but also elucidates how the liver function genes are marked only by H3K4me3 in this setting. Further, Figure S6D-E shows a chromatin state which is absolutely H3K4me- and H3K27me3+.*

2. Given that the bivalent marked cell cycle genes are activated during regeneration with reduction of H3K27me3, it is interesting to learn bivalent marked genes which are not related to cell cycle but to other hepatic regeneration. In addition, how many pro-regenerative genes are bivalent?

We appreciate this comment as it led to deeper insight into our data. We have indicated

the gene number in each cluster defined in **NEW Figure 5E** (total 584 genes), and the gene list is now provided in **NEW Table S3**. We also expanded the analysis of the GO pathways, which characterize each state, and highlight the cell cycle related genes, which are silenced and are in State 2 (**NEW Figure 4D and NEW Table S4**). To extend this, we surveyed the chromatin status of 157 genes annotated as functioning in the Hippo/Yap pathway, which is well studied in hepatic regeneration (Li et al. 2019; Pepe-Mooney et al. 2019). There are 121 genes fall into open chromatin

states, while 21 genes are bivalent. This suggests that in addition to the cell cycle regulators, genes in this key pathway may be poised for activation in the quiescent liver, consistent with findings by Li et al (2019).

3. The authors identified cell cycle genes were bivalent, which were pre-opened and marked with H3K27me3 and H3K4me3. Are these chromatin state and modifications resolved after PH? How about other chromatin features change at these genes during regeneration after PH?

*This is an interesting question which the current status of our data allows us to address partially, but a full profiling of the chromatin states during liver regeneration are beyond the scope of the current study. We have included new methylome data from 96 hours after PH which we incorporated with the H3K27me3 data previously shown at 0, 30, 40 and 96 hours. We show that H3K27me3 occupancy on promoters is close to being restored to baseline levels at 96 hours after PH, whereas there is very little difference in the level of DNA methylation (**NEW Figure S7C**) between 0 and 96 hours after PH. While profiling other marks would be advantageous for understanding the full scope of epigenetic changes during regeneration, these studies are ongoing and will be the focus of future work.*

4. The authors suggested that bivalent feature delivered marked genes a potential for regeneration. Permissive state was found in quiescent hepatocytes, which contributed to liver regeneration. For example, the study by Li et al (2019, Cell Stem Cell 25, 54-68) has already described a pre-existing accessible chromatin status in quiescent livers, which should be thoroughly discussed or compared in this paper.

We are very pleased that the reviewer suggested incorporating the findings of Li et al, as deeper investigation into this study led to a new finding. Li et al defined a geneset that was enriched in the liver progenitor like cells (LPLC) that emerge upon liver injury and showed that that the permissive state that allowed these genes to be expressed was dependent on Arid1a. We asked whether this LPLC geneset was encompassed by one or more of the chromatin states we described here. They report that a permissive epigenetic state allowed these genes to be expressed. We found that that 1155 of these 1881 LPLC genes resided in the predicted open chromatin states we defined (S1, S2 and S3) with the majority being encompassed by S2 (**NEW Figure S8D**), suggesting the pre-existing accessible chromatin status defined by Li et al. is similar to this chromatin state. Future work to define whether the chromatin states we defined here are shaped by Arid1a will be of interest.

5. Transcriptome after PH was analyzed in detail by Wang et al. (2019, Developmental Cell 50, 43–56). The distinct expression kinetics of genes may partially reflect their chromatin states. Could the authors discuss or perform additional characterizations of the chromatin state of these genes?

To address this interesting question, we analyzed the regenerative co-expressed gene clusters that we described in our previous publication (Wang et al, 2019). We first checked whether any of the genes falling into the co-expressing clusters are present in a ‘active’

chromatin states (S1-S2-S3) and then checked whether the cell cycle genes, which we previously identified as pro-regenerative and sensitive to changes in the repressive epigenome, were also enriched in any of the chromatin states defined in the current study. The result in Reviewer Figure 3A showed that the majority of the genes (72%; 5062 genes) that fall into the co-expression clusters (i.e. all clusters) are in S1 and S2, and only 10.44% of

these genes fall into the ‘repressive states’ S5 and 6. The result in Reviewer Figure 3B shows that over 80% of the Cluster 6 genes identified in Wang et al. as pro-regenerative, cell cycle genes are in S1 and S2, with only 1.2% of these genes in repressive states 5 and 6. While this is interesting and ties our previous study to this current work, we believe

that this is a minor point and therefore have provided the details here for the reviewer's benefit but did not elect to include it in what is already a very data-heavy manuscript.

6. An interesting finding is that the combination of H3K9me3 and DNA methylation marked the youngest transposons. Are there any special features of these TEs related to liver biology?

*It is of interest to the field to determine what TEs are relevant to liver biology however, we did not find that there were any obvious connections between the young TEs marked by H3K9me3 and DNA methylation and liver biology. We explored the nature of the TE expression and repression relationships in a new analysis and dataset. We first examined TE expression during liver regeneration and found very little changes in expression in the 6 time points we examined after PH (**NEW Figure S7A**). To examine this further, we subcategorized TEs into class and examined their differential expression at 96 hours after PH compared to baseline, which showed very little difference based on class (**NEW Figure S7B**). Other studies showed genome-wide transcriptional de-repression of TEs during replicative senescence in mouse liver (Van Meter et al. 2014). Another study showed that chromatin accessibility across TEs in the liver varies by strain and diet, and that this impacts TE expression (Du et al. 2016). Thus, the specific role of these differentially marked TEs is an open area for exploration.*

7. The authors classified chromatin status by chromHMM. However, there were some inconsistencies in the study. 1) the bivalent chromatin which was highlighted in this work was not found in six chromatin status in Figure 1A; 2) in Figure 5, ATAC, H3K4me3, H2AZ and H3K27me3 were detectable in S4, which was proposed as a low signal chromatin. Authors should provide analysis to explain these inconsistencies and showed a quality control of ChIP-seq data. (accumulated because broader regions)

We attribute what looks like inconsistencies to the different visualization methods used and address each point below.

1) The predicted 6 chromatin states in Figure 1 are based on probability, while the bivalency shown in Figure 5E was plotted with real Chip-seq signals. Figure 1 is intended as a broad view of the chromatin states of the entire hepatic genome; the bivalent aspects are a zoomed-in view of only specific marks and therefore provide more detailed analysis. An additional point is that the bivalent genes are grouped with other features using this process and since this is only a small fraction of all the loci profiled, this feature alone is insufficient to designate an entire state.

2) *While the appearance of multiple marks in the 'empty' state appears to be an inconsistency, it reflects the fact that S4 accounts almost 90% of genome; this means that signals are dispersed over this vast space and therefore are considered negligible. However, in Figure 4A we selected only the regions from S4 that have a positive signal. We did thorough QC on both our Chip-seq and ENCODE datasets and provide these in Supplemental Figures S1 and S2 to verify algorithm performance.*

Minor concerns

1. At Page 10, there is no Fig2F. Fig2E and Fig2F probably mean Fig2D and Fig2E.

We apologize for this mislabeling and have corrected in the current version.

2. It was read that “As expected, while all genes in S5 were covered with H3K27me3, very few also show occupancy by H3K4me3”. In Fig. 5a, no H3K27me3 enrichment in S5 was found.

We apologize for this mislabeling and have corrected in the current version.

3. In Fig3, the enrichment of genomic elements in ChromHMM states needs statistical analyses.

We have updated the text to address this. ChromHMM is a well-established method that depends on statistical evaluation to identify states (Ernst and Kellis 2017). ChromHMM takes qualified and normalized reads as a set of input and uses machine learning to define those chromatin-state signatures using a multivariate hidden Markov model to model the combinatorial presence or absence of each mark. When executed, ChromHMM learned a chromatin-state mode and assigned pieces of genome into different states (for which the state number is optimized by us) with the maximum possibility, for which statistical assessment of probability is integral. After learning the model, it was used to annotate the genome based on these combinatorial and further validated the model prediction with original Chip-seq signals to avoid false positive regions from modeling.

4, Fig5C implies that liver enriched genes almost absent from S3, S5 and S6. However, this figure highlights the orange points as liver enriched genes, which seem to be equally distributed in several classes. The authors may provide the proportion of liver enriched genes in S3, S5 and S6.

Figure 4C has been updated to include the number of liver enriched genes in each state.

References:

- Babaian, Artem, and Dixie L. Mager. 2016. "Endogenous Retroviral Promoter Exaptation in Human Cancer." *Mobile DNA* 7(1): 1–21.
- Bajic, Vladimir B. et al. 2006. "Mice and Men: Their Promoter Properties." *PLoS Genetics* 2(4): 614–26.
- Chen, Tianyi et al. 2020. "Single-Cell Omics Analysis Reveals Functional Diversification of Hepatocytes during Liver Regeneration." *JCI Insight* 5(22).
- Du, Juan et al. 2016. "Chromatin Variation Associated with Liver Metabolism Is Mediated by Transposable Elements." *Epigenetics and Chromatin* 9(1): 1–16.
- Ernst, Jason, and Manolis Kellis. 2017. "Chromatin-State Discovery and Genome Annotation with ChromHMM." *Nature Protocols* 12(12): 2478–92.
- Grindheim, Jessica Mae, Dario Nicetto, Greg Donahue, and Kenneth S. Zaret. 2019. "Polycomb Repressive Complex 2 Proteins EZH1 and EZH2 Regulate Timing of Postnatal Hepatocyte Maturation and Fibrosis by Repressing Genes With Euchromatic Promoters in Mice." *Gastroenterology* 156(6): 1834–48.
- Iakova, Polina, Samir S. Awad, and Nikolai A. Timchenko. 2003. "Aging Reduces Proliferative Capacities of Liver by Switching Pathways of C/EBP α Growth Arrest." *Cell* 113(4): 495–506.
- Li, Weiping et al. 2019. "A Homeostatic Arid1a-Dependent Permissive Chromatin State Licenses Hepatocyte Responsiveness to Liver-Injury-Associated YAP Signaling." *Cell Stem Cell* 25(1): 54-68.e5.
- Lynch, Magnus D. et al. 2012. "An Interspecies Analysis Reveals a Key Role for Unmethylated CpG Dinucleotides in Vertebrate Polycomb Complex Recruitment." *EMBO Journal* 31(2): 317–29. <http://dx.doi.org/10.1038/emboj.2011.399>.
- Van Meter, Michael et al. 2014. "SIRT6 Represses LINE1 Retrotransposons by Ribosylating KAP1 but This Repression Fails with Stress and Age." *Nature Communications* 5.
- van Mierlo, G., Gert Jan C. Veenstra, Michiel Vermeulen, and Hendrik Marks. 2019. "The Complexity of PRC2 Subcomplexes." *Trends in Cell Biology* 29(8): 660–71..
- Pepe-Mooney, Brian J. et al. 2019. "Single-Cell Analysis of the Liver Epithelium Reveals Dynamic Heterogeneity and an Essential Role for YAP in Homeostasis and Regeneration." *Cell Stem Cell* 25(1): 23-38.e8.
- Reddington, James P. et al. 2013. "Redistribution of H3K27me3 upon DNA Hypomethylation Results in De-Repression of Polycomb Target Genes." *Genome Biology* 14(3).
- Shukla, Ruchi et al. 2013. "Endogenous Retrotransposition Activates Oncogenic Pathways in Hepatocellular Carcinoma." *Cell* 153(1): 101–11.
- Sisu, Cristina et al. 2020. "Transcriptional Activity and Strain-Specific History of Mouse Pseudogenes." *Nature Communications* 11(1)..
- Weitzman, Jonathan B. 2002. "The Mouse Genome." *Genome Biology* 3(1): 1729–40.

REVIEWERS' COMMENTS

Reviewer #1 (Remarks to the Author):

The authors of the present manuscript have responded perfectly to all the comments made by the reviewers. They have given a very detailed explanation to all the doubts and comments raised and the corresponding experiments / changes have been carried out that significantly improve the manuscript. The work is considered suitable for publication.

Reviewer #2 (Remarks to the Author):

The authors have made extensive revisions to the manuscript to respond to reviewer critiques. The new data and analysis support the conclusions and I believe the manuscript is much improved. I do have a few remaining questions related to the new data/analysis presented in Figure 5.

1. It would be helpful if the authors could provide more detail as to how they clustered the genes in Figure 5E. It's not clear to this reviewer how that was done.

2. In the main text description of Figure 5E the authors describe Cluster 1 as "highly enriched" for H3K4me3, Cluster 2 as "enriched" for H3K4me3 downstream of the TSS and Cluster 3 as having "lower H3K4me3 occupancy" (first sentence on pg 18). However, in describing the chromatin profiles of Esf1 and Mcm2, the authors write that cluster 3 genes have high H3K27me3 and H3K4me3 in quiescent livers and then a depletion of H3K27me3 during regeneration. These seem to be contradictory statements.

3. Lastly, related to Figure 5E, it might be helpful for the authors to include aggregate plots of H3K27me3/H3K4me3 (as was done for Figure 5A) to more clearly see the differences.

Reviewer #3 (Remarks to the Author):

I only have two small issues regarding the data in Fig 5F. 1, The expression data/curves are difficult to see. Please enlarge. 2, It would be helpful to understand the data if a GO analysis is performed for the 3 clusters of genes.

REVIEWERS' COMMENTS

Reviewer #1 (Remarks to the Author):

The authors of the present manuscript have responded perfectly to all the comments made by the reviewers. They have given a very detailed explanation to all the doubts and comments raised and the corresponding experiments / changes have been carried out that significantly improve the manuscript.

The work is considered suitable for publication.

Reply: *We appreciate the reviewer's insightful comments as these helped to improve the manuscript significantly. We are grateful for the further endorsement.*

Reviewer #2 (Remarks to the Author):

The authors have made extensive revisions to the manuscript to respond to reviewer critiques. The new data and analysis support the conclusions and I believe the manuscript is much improved. I do have a few remaining questions related to the new data/analysis presented in Figure 5.

1. It would be helpful if the authors could provide more detail as to how they clustered the genes in Figure 5E. It's not clear to this reviewer how that was done.

Reply: *We have provided more details on how we clustered genes in this dataset as requested by the reviewer. We have now included further details about the method and parameters in both the results and material methods sections.*

2. In the main text description of Figure 5E the authors describe Cluster 1 as “highly enriched” for H3K4me3, Cluster 2 as “enriched” for H3K4me3 downstream of the TSS and Cluster 3 as having “lower H3K4me3 occupancy” (first sentence on pg 18). However, in describing the chromatin profiles of Esf1 and Mcm2, the authors write that cluster 3 genes have high H3K27me3 and H3K4me3 in quiescent livers and then a depletion of H3K27me3 during regeneration. These seem to be contradictory statements.

Reply: *We see how the wording describing Figure 5 could have been misleading. We meant to say that these genes have high H3K4me3 compared to those in the ‘closed’*

chromatin states. We corrected the text in the manuscript to indicate that these genes have 'high H3K27me3 and poised H3K4me3'.

3. Lastly, related to Figure 5E, it might be helpful for the authors to include aggregate plots of H3K27me3/H3K4me3 (as was done for Figure 5A) to more clearly see the differences.

Reply: *We appreciate this point helping make the figure clearer. We added aggregate plots above the heatmaps in Figure 5A.*

Reviewer #3 (Remarks to the Author):

I only have two small issues regarding the data in Fig 5F. 1, The expression data/curves are difficult to see. Please enlarge. 2, It would be helpful to understand the data if a GO analysis is performed for the 3 clusters of genes.

Reply: *We appreciate this points to improve the figure and modified it according to the reviewer's suggestions. We also did GO analysis for each of 3 clusters to show their specificities as NEW Supplementary Figure 6F.*